# UDC-VIX: A Real-World Video Dataset for Under-Display Cameras

## Abstract

Under Display Camera (UDC) is an advanced imaging system that places a digital camera lens underneath a display panel, effectively concealing the camera. However, the display panel significantly degrades captured images or videos, introducing low transmittance, blur, noise, and flare issues. Tackling such issues is challenging because of the complex degradation of UDCs, including diverse flare patterns. Despite extensive research on UDC images and their restoration models, studies on videos have yet to be significantly explored. While two UDC video datasets exist, they primarily focus on unrealistic or synthetic UDC degradation rather than real-world UDC degradation. In this paper, we propose a real-world UDC video dataset called UDC-VIX. Unlike existing datasets, only UDC-VIX exclusively includes human motions that target facial recognition. We propose a video-capturing system to simultaneously acquire non-degraded and UDC-degraded videos of the same scene. Then, we align a pair of captured videos frame by frame, using discrete Fourier transform (DFT). We compare UDC-VIX with seven representative UDC still image datasets and two existing UDC video datasets. Using six deep-learning models, we compare UDC-VIX and an existing synthetic UDC video dataset. The results indicate the ineffectiveness of models trained on earlier synthetic UDC video datasets, as they do not reflect the actual characteristics of UDC-degraded videos. We also demonstrate the importance of effective UDC restoration by evaluating face recognition accuracy concerning PSNR, SSIM, and LPIPS scores. UDC-VIX enables further exploration in the UDC video restoration and offers better insights into the challenge. UDC-VIX is available at our project site.

## 1 Introduction

An under-display camera (UDC) is an imaging system where the camera is positioned beneath the display (Hinton et al., 2006). Modern smartphones, including the Samsung Galaxy Z-Fold series (Samsung Electronics Co., Ltd., 2021; 2022; 2023) and the ZTE Axon series (ZTE Corporation, 2020; 2021; 2022) have adopted UDCs. The UDC area, depicted in Figure 1, serves as display space under normal circumstances and acts as the light's passage to the camera when capturing pictures or videos. This design allows for a larger screen-to-body ratio, meeting the common consumer demand for a full-screen display without a camera hole or notch. However, UDC introduces severe and complex image degradations such as reduced transmittance, noise, blur, and flare in a single image or video frame. Moreover, motion is also involved in UDC videos.

The degradation in UDC arises from the diffraction of incoming light by the display pixels at a micrometer scale (Qin et al., 2016). Modern UDC smartphones have lower pixel density in the UDC area to minimize this diffraction, as described in Figure 1(c). Since a lower pixel density prevents natural video viewing, improving the video quality captured by the UDC is essential.

Many studies have investigated UDC image datasets. These include synthetic datasets like T-OLED/P-OLED (Zhou et al., 2021) and SYNTH (Feng et al., 2021). Additionally, there exists a pseudo-real UDC dataset (Feng et al., 2023) and a real-world UDC dataset such as UDC-SIT (Ahn et al., 2024).

Ahn et al. (2024) demonstrate the importance of training DNN models using a real-world UDC dataset because the synthetic UDC datasets do not reflect the actual characteristics of UDC-degraded

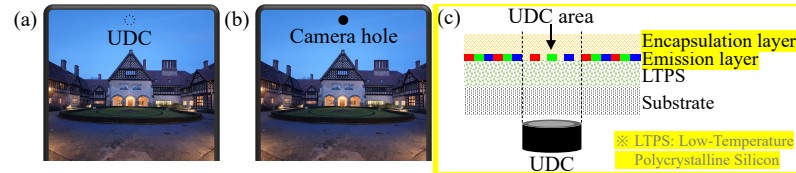

Figure 1: Comparison between under-display (UDC) and traditional hole-display cameras. (a) UDC. (b) Hole display camera. (c) The pixel structure of the UDC area. The UDC area exhibits a reduced pixel density due to the pixel pattern acting as diffraction slits.

images. However, a real-world UDC video dataset and restoration model have yet to be introduced. Although several studies address the synthetic UDC video datasets (Chen et al., 2023; Liu et al., 2024), they have some limitations because they do not completely reflect the properties of actual UDC videos. There are two main challenges in constructing a real-world UDC video dataset. One is to find a matching pair of the UDC-distorted and ground-truth videos with high alignment accuracy. The other is to synchronize the time for all frames when capturing videos.

This paper proposes a new UDC video dataset called UDC-VIX (**UDC**'s **VI**deo by **X**, where X represents the anonymous creator). As far as we know, it is the first real-world UDC video dataset to overcome the problems of the existing UDC video datasets.

Using a non-polarizing cube beam splitter (Thorlabs, 2015), we create a video-capturing system to minimize discrepancies between paired frames. We cut the UDC area of a smartphone display (e.g., Samsung Galaxy Z-Fold 5 (Samsung Electronics Co., Ltd., 2023)) and attach it to the beam splitter. Two Arducam Hawk-Eye (IMX686) camera modules (Arducam, 2022) are placed on both sides of the beam splitter. These modules, operated by a Raspberry Pi 5 (Arducam, 2023), capture synchronized video frame pairs using the `Message Passing Interface (MPI)` barrier.

Figure 2 shows our UDC video capturing system. Despite the meticulous design, inevitable pixel-position difference occurs. We correct this difference between the two matched frames for the same scene by using the DFT (Brigham, 1988) following the previous work by Ahn et al. (2024).

The contributions of this paper are summarized as follows:

- We address the limitations of existing datasets, including unrealistic degradations, improbable flares, and white artifacts, emphasizing the need for a high-quality, real-world dataset.
- We provide UDC-VIX, a real-world UDC video dataset that accurately reflects actual UDC degradations, ensuring precise spatial and temporal alignment through our meticulously designed video-capturing system.
- We describe UDC-VIX's effectiveness through extensive experiments, comparing it with an existing synthetic dataset using six deep-learning models. High-quality datasets and benchmarks are crucial for advancing representation learning.
- We highlight the importance of restoring UDC degradation for practical applications like Face ID by measuring face recognition accuracy at different restoration levels. Our dataset uniquely includes real-world face images, making it highly relevant for real-world tasks.

## 2 RELATED WORK

**Existing UDC image datasets.** There has been extensive research on UDC still image datasets. Zhou et al. (2021) propose the T-OLED/P-OLED datasets. Images are displayed on a monitor, and paired images are captured with and without a T-OLED/P-OLED display in front of the camera. However, due to the limited dynamic range of the monitor, flares are almost absent in their datasets. Feng et al. (2021) propose the SYNTH dataset. They convolve the measured point spread function (PSF) of ZTE Axon 20 (ZTE Corporation, 2020) with clean images (Haven, 2020), exhibiting flares. However, it has limitations such as the absence of noise and *spatially variant flares*. Notably, UDC distortion gradually increases from the center of the camera lens to outwards, leading to spatially distorted flares (Yoo et al., 2022). Feng et al. (2023) propose a pseudo-real dataset by capturing

paired images of similar scenes using two cameras (e.g., ZTE Axon 20 UDC (ZTE Corporation, 2020) and iPhone 13 Pro camera (Apple Inc., 2021)). However, they use two cameras, leading to geometric misalignment. They improve the geometric misalignment using AlignFormer (Feng et al., 2023). Nonetheless, they encounter challenges with alignment accuracy. Ahn et al. (2024) propose a real-world dataset called UDC-SIT and an image-capturing system. They attach Samsung Galaxy Z-Fold 3 (Samsung Electronics Co., Ltd., 2021)'s UDC area to a lid. Paired images are acquired by opening and closing the lid onto the Samsung Galaxy Note 10's standard camera (Samsung Electronics Co., Ltd., 2019). They use DFT to align the misalignment between the paired images that occurs during the opening and closing of the lid. The images in the UDC-SIT dataset contain the actual UDC degradation (e.g., *spatially variant flares*). Finally, Wang et al. (2024) and Tan et al. (2023) propose still image datasets for face recognition. However, these datasets are synthesized using a GAN-based model trained on the T/P-OLED dataset (Zhou et al., 2021), which lacks realistic UDC degradation, particularly flares. Moreover, the datasets are not publicly available.

**Existing UDC video datasets.** Research has been conducted on synthetic UDC video datasets. Chen et al. (2023) propose the PexelsUDC-T/P dataset. They train a GAN-based UDC video generation model using T-OLED/P-OLED datasets (Zhou et al., 2021), which do not show flares. They generate UDC-degraded videos using clean videos (Pexels, 2014). Moreover, the datasets are not publicly available. Liu et al. (2024) propose the VidUDC33K dataset. They convolve the measured PSF on the clean video frames (Haven, 2020) to show flares. They simulate the dynamic change of the PSF (Kwon et al., 2021) between consecutive frames following the previous work (Babbar & Bajaj, 2022; Liu et al., 2022a; Ye et al., 2021). However, flares in their dataset are unrealistic.

**UDC image restoration.** There has been active research on UDC image restoration. DISC-Net (Feng et al., 2021) incorporates the domain knowledge of the UDC image formation model. UDC-UNet, a second performer of MIPI challenge (Feng et al., 2022), introduces kernel branches to incorporate prior knowledge and condition branches for spatially variant manipulation.

**Video restoration.** Many studies have focused on video restoration models for general tasks, such as denoising (Tassano et al., 2020), deblurring (Wang et al., 2019; Zhong et al., 2020), and super-resolution (Wang et al., 2019). Unlike image restoration, which only focuses on a spatial dimension, video restoration leverages temporal information. FastDVDNet (Tassano et al., 2020) uses a two-step denoising process in a multi-scale architecture to leverage temporal information without explicit motion estimation. EDVR (Wang et al., 2019) aligns features using deformable convolutions (Dai et al., 2017) and applies both temporal and spatial attention to highlight essential features. ESTRNN (Zhong et al., 2020) integrates residual dense blocks into RNN cells for spatial feature extraction and employs a spatiotemporal attention module for feature fusion. However, studies on UDC video restoration are still rare. DDRNet (Liu et al., 2024), the pioneering work to address UDC video degradation, adopts a recurrent architecture that merges multi-scale feature learning and bi-directional propagation.

## 3 DATASET ACQUISITION

Since obtaining well-synchronized and precisely aligned paired videos for the same scene is challenging, we carefully design both hardware and software for capturing videos.

### 3.1 THE VIDEO CAPTURING SYSTEM

As shown in Figure 2, we present a UDC video capturing system consisting of two camera modules, a display panel for the UDC area, a beam splitter, two 6-axis stages, and a single-board computer. In this setup, one of the two camera modules is under low light conditions caused by the display panel, making synchronization between paired frames more challenging than in previous beam splitter setups (Hwang et al., 2015; Joze et al., 2020; Li et al., 2023; Rim et al., 2020). To capture synchronized videos for the same scene, we propose a UDC video-capturing system that ensures precise camera synchronization and accurate frame alignment.

**The camera module.** We use the Hawk-Eye (IMX686) (Arducam, 2022) to ensure that UDC-VIX exhibits a similar UDC degradation as Samsung Galaxy Z-Fold 5's UDC. Both devices use *Quad*

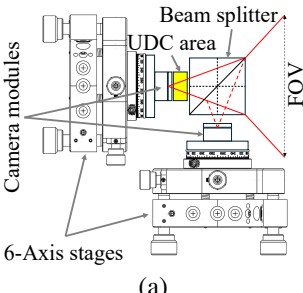 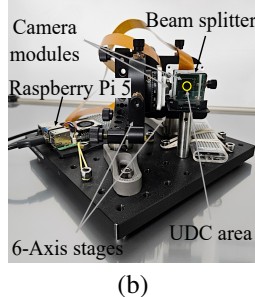

(a)                 (b)

Figure 2: The UDC video-capturing system. (a) The optical layout of the dual camera combiner. The UDC area is enlarged for a better view. (b) The proposed video-capturing system.

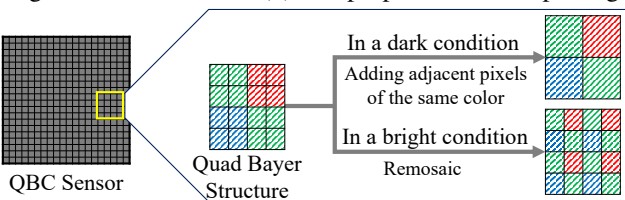

Figure 3: Quad Bayer Coding (QBC).

*Bayer Coding (QBC)*, a technique designed to mitigate the reduced sensor luminance sensitivity often associated with higher camera resolutions or smaller sensor pixel sizes (Sony, 2014; Wikipedia, 2014). As shown in Figure 3, four adjacent pixels share the same color filter in the quad Bayer structure. These pixels are grouped to increase sensitivity and reduce noise in low-light conditions (e.g., in the UDC setting). Conversely, in bright conditions, the sensor reverts the pixels to the Bayer structure through the remosaicing process, maintaining the Bayer sensor's high resolution.

**The beam splitter.** We use a non-polarizing cube-shaped beam splitter (e.g., Thorlabs CCM1-BS013 (Thorlabs, 2015)) to enable the two camera modules to capture the same scene. The beam splitter divides the incident light into two directions with a beam deviation of $0 \pm 5$ arcminutes at a 50:50 ratio. The unused optical path of the combiner is black-coated to minimize image contrast loss caused by scattering. They can capture the same scene by aligning the two cameras to the beam splitter's split fields of view (FOV). Figure 2(a) illustrates the optomechanical layout of the dual camera and beam splitter.

**Kinematic optical mount.** ==Despite many studies using beam splitters for paired image dataset collection (Hwang et al., 2015; Joze et al., 2020; Li et al., 2023; Rim et al., 2020), this paper is the first to apply them in UDC research to the best of our knowledge.== It presents challenges to align the optical paths of the display panel's UDC area, a beam splitter, and two camera modules. To ensure alignment between the cameras' optical axes and the beam splitter, we employ Thorlabs K6XS 6-axis kinematic optical mounts (Thorlabs, 2013). Each camera module is mounted on a K6XS mount, allowing for shifts, rotations, and tilts across the six axes to align their FOV.

**The controller.** We use a Raspberry Pi 5 (Arducam, 2023) that has two four-lane MIPI interface connections for high bandwidth to synchronize the two high-resolution cameras. It ensures stable high-resolution video recording. To synchronize the cameras, we use independent streamers managed by MPI barriers (Message Passing Interface Forum, 2023), achieving synchronization with an accuracy margin of up to 8 msec. Consistent frame rates for both cameras are ensured using the uncompressed binary dump method (e.g., `YUV420` format). Despite these settings resulting in less than a 0.5 fps (8 msec) difference between paired frames, rapid movements may still cause the cameras to capture different scenes. Thus, videos capturing fast-moving objects (e.g., speeding cars) are excluded from the dataset.

### 3.2 OBTAINING ALIGNED VIDEO PAIRS

This section illustrates how we align the optical axis and FOV, the criteria for determining FOV alignment, and the test cases. We use a real-time monitor viewing system for the two cameras. We

roughly align the view, fine-tuning using the K6XS, DFT alignment, the accuracy evaluation (e.g., PCK), video recording, and final selection by humans. The specific algorithm for the alignment is described in Algorithm 1. Please see Section 4 for detailed information on the PCK.

---

**Algorithm 1** Aligned video capturing process for UDC-VIX.

---

**Ensure:** The alignment accuracy of the paired frames is greater than 90%.
   **while** The average PCK ($\alpha = 0.005$) < 90% **do**
      **Initial setup.** Adjust the camera positions and the beam splitter, ensuring that the views of the two cameras are roughly similar.
      **Fine-tuning.** The rotation, tilt, and horizontal/vertical positions of the K6XS are finely adjusted by observing a $12 \times 9$ checkerboard and everyday scenes in the live view system.
      **DFT alignment and PCK evaluation.** Align paired frames using DFT and calculate the average PCK.
   **end while**
   **Video recording.** Capture paired videos for the same scene.
   **Final selection.** Only the videos all authors assessed aligned and synchronized are retained.

---

**DFT alignment.** Despite the careful design of our video-capturing system, unavoidable misalignments, such as *shifts*, *rotations*, and *tilts*, still occur between paired frames. Previous methods, such as SIFT (Lowe, 2004), RANSAC (Fischler & Bolles, 1981), and deep learning approaches (Feng et al., 2023), struggle to perform well in the existence of severe degradation introduced by the UDC. Thus, we use DFT to align the paired frames, following Ahn et al. (2024)'s alignment technique to achieve degradation-resilient alignment.

The alignment process is summarized as *shift, rotate, and crop* paired frames using DFT. Captured videos have an original frame size of $(1920, 1080, 3)$. The ground-truth frame is center-cropped to $(1900, 1060, 3)$, and the degraded frame undergoes a cropping around the center. To align the cropped degraded frame $\mathcal{D}$ with the cropped ground-truth frame $\mathcal{G}$, we iteratively *shift* the $(x, y)$ coordinates and *rotate* the frames to find the point of minimum loss. Our focus is on addressing *shifts* and *rotations* while excluding *tilts*. Handling *tilts* is challenging because of the need for perspective transforms optimized for objects in the same plane within a single image. Despite not considering *tilts*, our video-capturing system minimizes all *shifts*, *rotations*, and *tilts* so that they do not significantly affect alignment, as confirmed by our experiment (the PCK values in Table 2). The loss function $\mathcal{L}$ for the alignment between $\mathcal{D}$ and $\mathcal{G}$ is defined as below:

$$\mathcal{L} = \lambda_1 \sum_{x=0}^{M-1} \sum_{y=0}^{N-1} (\mathcal{D}(x,y) - \mathcal{G}(x,y))^2 + \lambda_2 \sum_{u=0}^{M-1} \sum_{v=0}^{N-1} \Delta\mathcal{F}_{amp}(u,v) + \lambda_3 \sum_{u=0}^{M-1} \sum_{v=0}^{N-1} \Delta\phi(u,v), \quad (1)$$

where the first term is the mean squared error, $\Delta\mathcal{F}_{amp}(u,v)$ and $\Delta\phi(u,v)$ represent the L1 distance for the amplitude and phase, respectively. They are defined as $\Delta\mathcal{F}_{amp}(u,v) = |\mathcal{F}_{\mathcal{D}}(u,v) - \mathcal{F}_{\mathcal{G}}(u,v)|$ and $\Delta\phi(u,v) = |\phi_{\mathcal{D}}(u,v) - \phi_{\mathcal{G}}(u,v)|$. Note that $\mathcal{F}(u,v)$ is the frequency value at the point $(u,v)$ in the frequency domain. Following Ahn et al.'s setting, we use $\lambda_1 = \lambda_3 = 1, \lambda_2 = 0$. The detailed alignment algorithm is described in the supplementary material.

## 4 COMPARISON WITH THE EXISTING UDC DATASETS

Many synthetic UDC datasets, including VidUDC33K (Liu et al., 2024), formulate the UDC degradation as follows:

$$I_t^D = f(\gamma \cdot I_t^G * k_t + n), \quad (2)$$

where $I_t^D$ and $I_t^G$ denote the UDC-degraded and ground-truth frames, respectively. $\gamma$ is the intensity scaling factor, $k_t$ refers to the diffraction kernel (i.e., PSF), $n$ is the noise, and $f$ denotes the clamp function for the pixel value saturation.

Ideally, we would like to compare UDC-VIX with two existing UDC video datasets, PexelsUDC (Chen et al., 2023) and VidUDC33K (Liu et al., 2024). However, since PexelsUDC is not publicly available, we use the P-OLED dataset (Zhou et al., 2021) used to create it. Table 1 gives a summary of the nine previous UDC datasets. The resolution and frame per second (`fps`) of UDC-VIX are FHD and 60 `fps`, respectively, following the Samsung Galaxy Z-Fold 5's specification.

Table 1: Comparison of the UDC datasets. The dataset size refers to the number of images in the image dataset or the total number of frames in the video dataset, calculated as the product of the number of video clips and the number of frames per clip. For example, the UDC-VIX dataset consists of 647 video clips with 180 frames per clip, so the total number of frames is 116,460.

| Dataset | Type | Scene | Dataset size | Resolution | fps | Flare presence | Face recognition | Publicly available | Publication |
|---|---|---|---|---|---|---|---|---|---|
| T/P-OLED (Zhou et al., 2021) | Image | Synthetic | 300 | $1024 \times 2048 \times 3$ | - | | | ✔ | CVPR '21 |
| SYNTH (Feng et al., 2021) | Image | Synthetic | 2,376 | $800 \times 800 \times 3$ | - | ✔ | | ✔ | CVPR '21 |
| Yoo et al. (Yoo et al., 2022) | Image | Synthetic | - | - | - | ✔ | | | SID '22 |
| Pseudo-real (Feng et al., 2023) | Image | **Real** | 6,747 | $512 \times 512 \times 3$ | - | ✔ | | ✔ | CVPR '23 |
| UDC-SIT (Ahn et al., 2024) | Image | **Real** | 2,340 | $1792 \times 1280 \times 4$ | - | ✔ | | ✔ | NeurIPS '23 |
| Tan et al. (2023) | Image | Synthetic | 73,000 | - | - | | ✔ | | TCSVT '23 |
| Wang et al. (2024) | Image | Synthetic | 56,126 | - | - | | ✔ | | arXiv '24 |
| PexelsUDC-T/P (Chen et al., 2023) | **Video** | Synthetic | $160 \times 100$ (16,000) | $1280 \times 720 \times 3$ | 25-50 | | | | arXiv '23 |
| VidUDC33K (Liu et al., 2024) | **Video** | Synthetic | $677 \times 50$ (33,850) | $1920 \times 1080 \times 3$ | - | ✔ | | ✔ | AAAI '24 |
| UDC-VIX | **Video** | **Real** | $647 \times 180$ (116,460) | $1900 \times 1060 \times 3$ | 60 | ✔ | ✔ | ✔ | |

**Noise and transmittance decrease.** The camera sensor amplifies the desired signal and unwanted noise in low-light conditions. In the UDC setting, where the sensor is beneath the display panel, the transmittance decreases, leading to amplified noise. The camera sensors with QBC, used in the Samsung Galaxy Z-Fold series (related to UDC-VIX) (Samsung Electronics Co., Ltd., 2021; 2022; 2023) and ZTE Axon series (related to VidUDC33K) (ZTE Corporation, 2020; 2021; 2022), can influence the noise pattern and pixel intensity (Sony, 2014). Thus, adding noise and adjusting intensity scaling values in Equation 2 may not accurately depict real-world noise and transmittance reduction. For example, in the VidUDC33K dataset, the degraded frame's noise level is somewhat lower than the ground truth, as shown in Figure 4(b). Similarly, the P-OLED dataset, captured in a controlled setting, exhibits unrealistic noise and excessive transmittance decrease, as depicted in Figure 4(a). In contrast, UDC-VIX in Figure 4(c) accurately shows actual transmittance decrease and digital noise resulting from quantizing digital image signals.

**Flares.** Conventional lens flares stem from intense light scattering or reflection within an optical system (Dai et al., 2022; 2023). In contrast, UDC flares arise from light diffraction as it passes through the display panel above the digital camera lens. Thus, it is crucial for each frame in the UDC video dataset to precisely depict the its unique flare characteristics, including *spatially variant flares*, *light source variant flares*, and *temporally variant flares*. The P-OLED dataset rarely exhibits flares as it captures images displayed on a monitor in a controlled environment (Figure 5(a) and (d)).

Since UDC distortion increases outward from the camera lens center, *spatially variant flares* manifest within an image (Yoo et al., 2022). Distorted PSFs must be convolved across different image regions to depict this flare distortion accurately. However, VidUDC33K applies the same PSF convolution across all areas using Equation 2, failing to represent spatially variant flares, as illustrated in Figure 5(b) and (e). Conversely, UDC-VIX effectively captures spatially variant flares (Figure 5(c)).

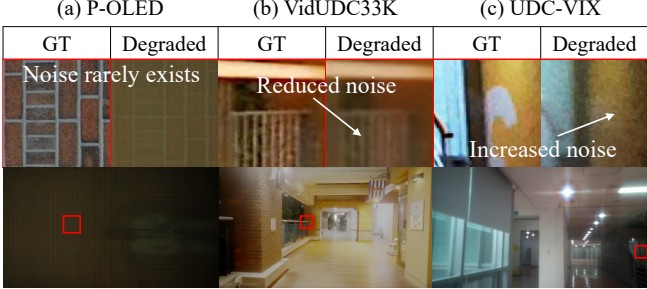

Figure 4: Comparison of the decrease in transmittance and digital noise by the UDC. (a) P-OLED dataset rarely depicts noise. (b) In the VidUDC33K dataset, the degraded frame decreases digital noise compared to the ground truth (GT) frame. (c) UDC-VIX dataset illustrates an increase in digital noise in the degraded frame. The brightness has been adjusted to improve visibility.

Figure 5: Comparison of flares. P-OLED shows no flares ((a) and (d)). VidUDC33K displays overly regular flares and light source invariant flares ((b) and (e)). In contrast, UDC-VIX uniquely presents *spatially variant flares* and *light source variant flares* ((c) and (f)).

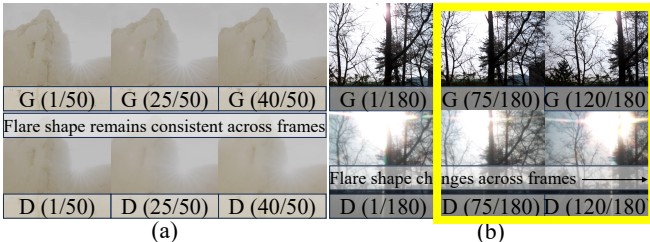

Figure 6: Temporally variant flares. Unlike (a) VidUDC33K, (b) UDC-VIX shows temporally variant flares. G and D are the ground truth and degraded frames, respectively. The numbers in parentheses represent (`the current frame number / the total number of frames`).

Various light sources, such as artificial (e.g., LED and halogen) and natural light, can alter the spectra, affecting UDC flares' shapes. However, VidUDC33K fails to depict *light source variant flares*. As seen in Figure 5(b) and (e), flare shapes remain similar despite different light sources. Conversely, UDC-VIX exhibits diverse flare shapes, as shown in Figure 5(c) and (f) and Figure 6(b).

A notable characteristic of UDC videos is *temporally variant flares* caused by the camera's motion when capturing light sources. The motion results in changes in PSFs (Kwon et al., 2021). However, in the VidUDC33K dataset, attempts to simulate PSF changes through inter-frame homography matrix computations using the method proposed by the previous studies (Babbar & Bajaj, 2022; Liu et al., 2022a; Ye et al., 2021) yield rare *temporally variant flares*, as shown in Figure 6(a). Moreover, the shape of typical lens flares in ground-truth frames remains unchanged in degraded frames, indicating the failure of PSF convolution to replicate natural sunlight flares. Conversely, UDC-VIX effectively captures temporally varying flares (Figure 6(b)).

**Face recognition.** UDC-VIX stands out from other datasets in Table 1 by featuring videos tailored for face recognition (FR). Some datasets, such as T-OLED/P-OLED, SYNTH, and VidUDC33K, only include limited human representations, often too small or from unrecognizable angles for FR (Figure 7(f)). Wang et al. (2024) introduce still image datasets for FR. However, these datasets are generated using a GAN-based model trained on the P-OLED dataset (Zhou et al., 2021), which does not adequately simulate realistic UDC degradation, notably the lack of flare (Figure 7(e)). Additionally, these datasets are not publicly available. Conversely, UDC-VIX prominently features humans in 64.6% of its videos (approved by the Institutional Review Board (IRB)), featuring various mo-

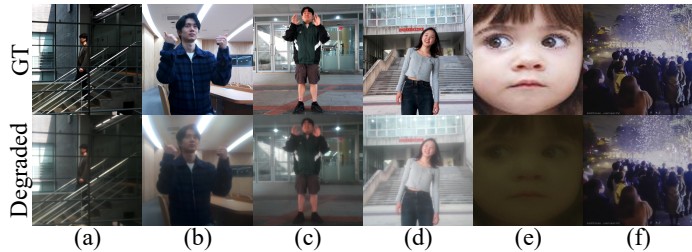

Figure 7: UDC-VIX features human motions, including (a) walking, (b) thumbs-up, (c) hand waving, and (d) body swaying. In contrast, Wang et al. (2024)'s synthetic still image datasets for FR do not show the actual UDC degradations, as shown in (e). Moreover, it is not publicly available. VidUDC33K dataset includes humans but is limited to rear views, as shown in (f).

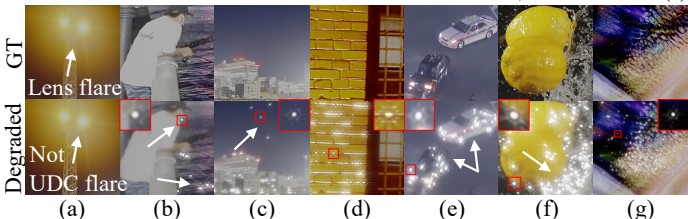

Figure 8: Less meaningful and strange videos in VidUDC33K (Liu et al., 2024). (a) Only lens flare is present, excluding UDC flare. (b) Flare in the seawater and on the cigarette. (c) Flare in the stars of the sky. (d) Flare on the bricks. (e) Flare on the car's side. (f) Flare on the splashing water droplets. (g) Meaningless abstract image.

tions (e.g., hand waving, thumbs-up, body-swaying, and walking) by 22 carefully selected subjects from different angles (Figure 7(a)-(d)).

**Less meaningful and strange scenes.** The VidUDC33K dataset often presents unrealistic scenarios. As depicted in Figure 8(a), degraded frames lack UDC flares, displaying flares resembling typical lens flares seen in the ground truth frame. Additionally, flares appear in improbable situations in Figure 8(b), (c), (d), (e), and (f). Moreover, some videos in VidUDC33K may not significantly contribute to research, prompting consideration for their relevance, as shown in Figure 8(g). Please see Appendix B.1 for detailed illustration.

**Alignment quality.** To assess the alignment quality of paired videos, we use LoFTR (Sun et al., 2021) as a keypoint matcher, following the convention of the previous studies (Ahn et al., 2024; Feng et al., 2023). We compare the Percentage of Correct Keypoints (PCK), representing the ratio of correctly aligned keypoints to the total number. A keypoint pair is correctly aligned if $d < \alpha \times max(H, W)$, where $d$ is the positional difference between a pair of matched keypoints, $\alpha$ is the threshold, and $H$ and $W$ are the frame or image dimensions. We set $max(H, W) = 1024$ for fair comparison across datasets with varying resolutions.

Table 2 compares alignment accuracy across datasets. The synthetic datasets (e.g., T-OLED/P-OLED, SYNTH, and VidUDC33K) do not require an additional alignment process, leading to PCK values near 100%. In contrast, the Pseudo-real dataset using AlignFormer (Feng et al., 2023), attains a PCK value of 58.75% for $\alpha = 0.01$. Unlike Pseudo-real, UDC-VIX maintains PCK values near 100%, demonstrating performance comparable to UDC-SIT, which previously led benchmarks.

## 5 EXPERIMENTS

This section compares the UDC video restoration performance and face recognition accuracy of the existing deep learning models trained by UDC-VIX and the existing synthetic video dataset.

### 5.1 EFFECTS ON LEARNABLE RESTORATION MODELS

In this section, we evaluate the effectiveness of the UDC-VIX dataset by comparing the video restoration performance of six deep learning models on UDC-VIX and VidUDC33K dataset (Liu

Table 2: The comparison of PCK values between the datasets. The UDC-VIX dataset showcases the best alignment quality. It has PCK values close to 100% for all values of $\alpha$.

| Dataset | Type | Need alignment | PCK ($\alpha = 0.01$) | PCK ($\alpha = 0.03$) | PCK ($\alpha = 0.10$) |
|---|---|---|---|---|---|
| T-OLED/P-OLED (Zhou et al., 2021) | Image | | 98.11 | 98.45 | 99.08 |
| SYNTH (Feng et al., 2021) | Image | | 99.95 | 99.96 | 99.99 |
| Pseudo-real (Feng et al., 2023) | Image | ✔ | **58.75** | **95.08** | **99.93** |
| UDC-SIT (Ahn et al., 2024) | Image | ✔ | **97.26** | **98.56** | **99.35** |
| VidUDC33K (Liu et al., 2024) | Video | | 99.82 | 99.84 | 99.90 |
| UDC-VIX | Video | ✔ | **98.95** | **99.32** | **99.69** |

Table 3: Restoration performance for synthetic and real UDC video datasets. The term *Input* refers to the PSNR, SSIM, and LPIPS values between the degraded and ground-truth video pairs.

| | Runtime (sec) | Param (M) | VidUDC33K (Liu et al., 2024) PSNR ↑ | SSIM ↑ | LPIPS ↓ | UDC-VIX PSNR ↑ | SSIM ↑ | LPIPS ↓ |
|---|---|---|---|---|---|---|---|---|
| Input | - | - | **26.22** | **0.8524** | **0.2642** | **16.31** | **0.7318** | **0.4165** |
| DISCNet (Feng et al., 2021) | 0.73 | 3.80 | 28.89 | 0.8405 | 0.2432 | 24.53 | 0.8351 | 0.2702 |
| UDC-UNet (Liu et al., 2023) | 0.37 | 5.70 | 28.37 | 0.8361 | 0.2561 | 27.74 | 0.8852 | 0.1814 |
| FastDVDNet (Tassano et al., 2020) | 0.45 | 2.48 | 28.95 | 0.8638 | 0.2203 | 23.76 | 0.8388 | 0.2696 |
| EDVR (Wang et al., 2019) | 1.17 | 23.6 | 28.71 | 0.8531 | 0.2416 | 23.40 | 0.8280 | 0.2700 |
| ESTRNN (Zhong et al., 2020) | 0.20 | 2.47 | 29.54 | 0.8744 | 0.2170 | 25.18 | 0.8599 | 0.2251 |
| DDRNet (Liu et al., 2024) | 0.44 | 5.76 | 31.91 | 0.9313 | 0.1306 | 24.49 | 0.8484 | 0.2255 |

et al., 2024). The comparison is performed only with VidUDC33K since PexelsUDC is not publicly available. DDRNet (Liu et al., 2024) is the only existing UDC video restoration model, while Fast-DVDNet (Tassano et al., 2020), EDVR (Wang et al., 2019), and ESTRNN (Zhong et al., 2020) are video restoration models for other general tasks (e.g., deblur, denoising, and super-resolution). DIS-CNet (Feng et al., 2021) and UDC-UNet (Liu et al., 2023) are UDC still image restoration models.

Table 3 shows the restoration performance of the six models on both VidUDC33K and UDC-VIX. Interestingly, the performance rankings of the benchmark models across the two datasets do not consistently align. The varying severity of flares between the two datasets is the main reason for the inconsistent restoration performance rankings. Unlike UDC-VIX, VidUDC33K lacks accurate depictions of real-world flares. Examination of input PSNR, SSIM, and LPIPS metrics indicates that their performance degradation on UDC-VIX is more severe than on VidUDC33K. The top performers on UDC-VIX, UDC-UNet and ESTRNN, use residual CNNs to manage complex degradations and enhance restoration quality. They also provide better frame-to-frame consistency than the others, which is crucial for reducing flicker, although some flicker persists. This shows the benefits of residual connections in improving consistency. Note that the restored video of VidUDC33K by DDRNet using their pre-trained model does not create flickering. This result underscores the necessity for research dedicated to UDC's video restoration using real-world UDC video datasets, an area where UDC-VIX holds promise for significant contributions. Extensive analyses and visual comparisons are available in Appendix B.2 and B.3, and on our project site.

## 5.2 FACE RECOGNITION

The face recognition (FR) task verifies whether two images are of the same person, similar to typical smartphone applications like Face ID. As shown in Figure 9, we assess average FR accuracy using seven FR models from the DeepFace library (Serengil, 2022), such as VGG-Face (Parkhi et al., 2015), Facenet (Schroff et al., 2015), OpenFace (Baltrušaitis et al., 2016), DeepFace (Taigman et al., 2014), DeepID (Sun et al., 2014), Dlib (King, 2009), and ArcFace (Deng et al., 2019). We test 600 FR frame pairs (human 1 and human 2 from different videos) on a balanced dataset, with 49.2% of the same person (human 1 = human 2) and 50.8% of different people (human 1 ≠ human 2).

As shown in Figure 9, we compare the effect of human 2's restoration level in terms of PSNR, SSIM, and LPIPS ($X$-axis) on FR accuracy ($Y$-axis). Human 1 is always ground truth (GT) and human 2 can be Input, Restored, or GT). Therefore, Input, Restored, or GT in Figure 9 indicates the group to which human 2 belongs. For example, in Figure 9(a), the PSNR for Input is calculated

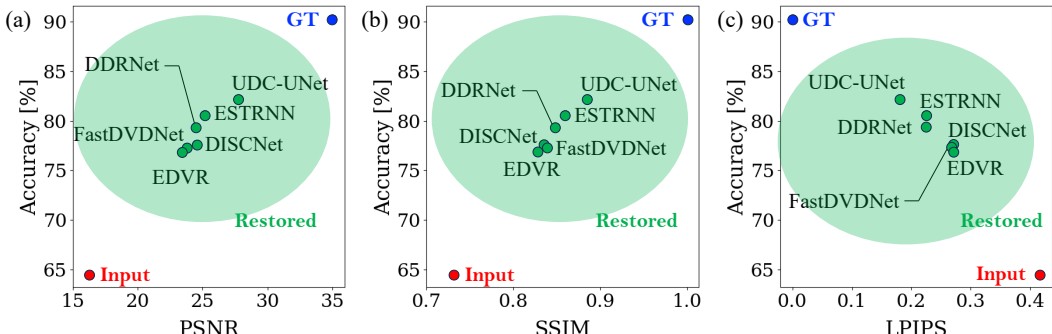

Figure 9: FR accuracy. Model error cases are excluded when calculating the accuracy, where the model error indicates when FR models fail due to severe UDC degradation. Frames restored by deep learning models with higher performance in (a) PSNR, (b) SSIM, and (c) LPIPS achieve better recognition accuracy. PSNR between the two GTs is plotted as 35.00 for easy observation.

between human 2 (`Input`) and human 2 (GT). Similarly, the FR accuracy for `Input` is calculated between human 1 (GT) and human 2 (`Input`). To verify the relationship between restoration level and FR accuracy, we illustrate six deep-learning models' restoration performance (highlighted with green circle) and corresponding FR accuracy in Figure 9. The PSNR for `Restored` is calculated between human 2 (`Restored`) and human 2 (GT). Similarly, the FR accuracy for `Restored` is calculated between human 1 (GT) and human 2 (`Restored`).

The results show the significance of leveraging the UDC degradation by deep-learning restoration models to enhance FR accuracy. For example, as depicted in Figure 9(a), `Input` with PSNR of 16.31 shows 64.5% FR accuracy, `UDC-UNet` with PSNR of 27.74 shows 82.2% FR accuracy, and `GT` shows 90.3% FR accuracy.

## 6 LIMITATIONS

UDC-VIX has two limitations. One is that UDC degradations vary with display pixel design, affecting diffraction patterns, PSF, and light propagation, leading to variation in degradation such as blur, transmittance decrease, and especially flares (see Figure 5(b) and (c)). Models trained on UDC-VIX may not work optimally on devices other than Samsung Galaxy Z-Fold 5 (Samsung Electronics Co., Ltd., 2023), such as the ZTE Axon series (ZTE Corporation, 2020; 2021; 2022) or other Samsung Galaxy Z-Fold series (Samsung Electronics Co., Ltd., 2021; 2022). However, models trained on UDC-VIX can be fine-tuned for other devices. Please see Appendix B.4 for details. The other is that fast-moving objects like speeding cars are excluded from UDC-VIX. Despite the efforts to synchronize the two cameras to ensure a synchronization difference of less than 8 msec between paired frames (Section 3), rapid movements can still result in scene difference by the cameras.

## 7 CONCLUSION

As far as we know, UDC-VIX is the first UDC video dataset that includes actual UDC degradation, such as low transmittance, blur, noise, and flare. We propose an efficient video-capturing system to acquire a matched pair of UDC-degraded and ground-truth videos with precise synchronization of two cameras. Furthermore, we align UDC-VIX frame by frame using DFT, showing the highest alignment accuracy, enough to train deep learning models. From the comparison experiments, we demonstrate the effectiveness of UDC-VIX. Notably, UDC-VIX solely presents significant actual UDC degradation (e.g., *variant flares*) and stands out from other datasets by featuring videos tailored for face recognition. Through the thorough experiments, we figure out the models trained with the synthetic UDC video dataset are impractical because they fail to capture UDC-degraded videos' actual characteristics accurately. Moreover, restoring UDC degradation is significant in enhancing face recognition accuracy. Based on the insights above, we expect that UDC-VIX will significantly contribute to the UDC video restoration studies.

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

# A DETAILS OF THE UDC-VIX DATASET

In this section, we provide detailed information about the UDC-VIX dataset.

## A.1 DATASET ACQUISITION

As described in Section 3 of the main body of the paper, to construct a real-world UDC video dataset with precise alignment and synchronization, we propose a video-capturing system. This section details the alignment algorithm based on Discrete Fourier Transform (DFT) and its advantages. We also describe the techniques for synchronized video capture using the two camera modules.

**Alignment.** The alignment algorithm we use involves *shifting*, *rotating*, and *cropping* paired frames with DFT. The detailed alignment algorithm is illustrated in Algorithm A.1. In this algorithm, following the alignment settings by Ahn et al. (2024), we use $\lambda_1 = \lambda_3 = 1$ and $\lambda_2 = 0$, and we do not apply rotation. Their experiments show that applying rotation reduces the Percentage of Correct Keypoints (PCK) when varying $\lambda_1$, $\lambda_2$, $\lambda_3$, and $\theta_{\text{rotation}}$.

The loss function in Equation 1 in the main body of the paper enables the incorporation of both local (i.e., MSE) and global (i.e., DFT) information across spatial and frequency domains. Using DFT to align the paired frames offers a significant advantage because it can decompose a frame into its constituent spatial frequency components. Figure A.1(a) and (c) depict paired frames $\mathcal{G}$ and $\mathcal{D}$ comprising multiple sinusoidal gratings, indicating a noticeable spatial shift. Figure A.1(b) and (d) represent the differences in phase and amplitude, respectively. Thus, reducing the phase component is critical for effectively aligning the paired frames for the same scene.

**The controller.** When capturing videos, we discard the initial 30 frames because it takes approximately 15 frames for the ground-truth camera and 25 frames for the UDC to achieve focus. The UDC requires more frames for focusing due to its degradation. Furthermore, we use a solid-state drive

---

**Algorithm A.1** Alignment of paired images $I_G$ and $I_D$ (Ahn et al., 2024).

---

**Require:** Images $I_G$, $I_D$ of size $(H, W)$, hyperparameters $s$, $\theta_r$, $r$, $\lambda_1$, $\lambda_2$, $\lambda_3$
**Ensure:** Aligned images $\mathcal{G}$, $\mathcal{D}$ of size $(H^*, W^*)$
    Crop $\mathcal{G}$ from $I_G$ using center crop
    Crop $\mathcal{D}$ from $I_D$ to the size of $\mathcal{G}$
    Initialize best loss $\mathcal{L}_{\text{best}}$ to a large value
    Initialize optimal shifts $s_{\text{opt\_x}}$, $s_{\text{opt\_y}}$, and rotation $\theta_{\text{opt}}$ to 0
    **for** $\theta_{\text{rotation}}$ from $-\theta_r$ to $\theta_r$ with step $r$ **do**
        Apply rotation of $\theta_{\text{rotation}}$ to $I_D$ to get $\mathcal{D}_{\text{rotated}}$
        **for** $x_{\text{shift}}$ from $-s$ to $s$ with step 1 **do**
            **for** $y_{\text{shift}}$ from $-s$ to $s$ with step 1 **do**
                Calculate crop position $(p, q)$ relative to the center crop:
                    $p = x_{\text{center\_crop}} + x_{\text{shift}}$
                    $q = y_{\text{center\_crop}} + y_{\text{shift}}$
                Crop image $\mathcal{D}_{\text{tmp}}$ from $\mathcal{D}_{\text{rotated}}$ at position $(p, q)$
                Calculate loss $\mathcal{L}$ using the loss function in **Eq. 1** between $\mathcal{D}_{\text{tmp}}$ and $\mathcal{G}$
                **if** $L < \mathcal{L}_{\text{best}}$ **then**
                    Update $\mathcal{L}_{\text{best}}$ to $L$
                    Update $s_{\text{opt\_x}}$ to $x_{\text{shift}}$
                    Update $s_{\text{opt\_y}}$ to $y_{\text{shift}}$
                    Update $\theta_{\text{opt}}$ to $\theta_{\text{rotation}}$
                **end if**
            **end for**
        **end for**
    **end for**
    Apply optimal rotation $\theta_{\text{opt}}$ to $I_D$ to get $\mathcal{D}_{\text{rotated}}$
    Calculate crop position $(p_{\text{opt}}, q_{\text{opt}})$ relative to the center crop:
        $p_{\text{opt}} = x_{\text{center\_crop}} + s_{\text{opt\_x}}$
        $q_{\text{opt}} = y_{\text{center\_crop}} + s_{\text{opt\_y}}$
    Crop $\mathcal{D}_{\text{rotated}}$ to acquire an aligned image $\mathcal{D}$ at position $(p_{\text{opt}}, q_{\text{opt}})$

---

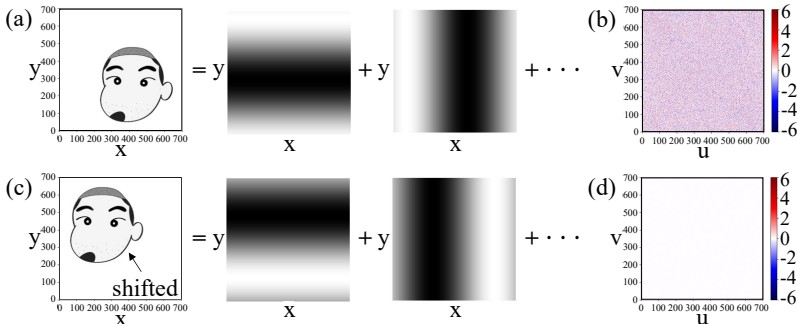

Figure A.1: Frequency analysis based on the conceptual illustration for paired frames involving shifts without degradation. (a) The original frame $\mathcal{G}$ consists of multiple sinusoidal gratings. The inverse $DFT$ applied to $\mathcal{F}_G(u, v)$ produces each sinusoidal grating. (b) The phase difference between $\mathcal{G}$ and $\mathcal{D}$. (c) The spatially shifted frame $\mathcal{D}$ in the spatial domain comprises multiple sinusoidal gratings, as in (a). (d) The amplitude difference between $\mathcal{G}$ and $\mathcal{D}$, showing no difference.

(SSD) instead of a secured digital (SD) card, as the SD card takes longer to save FHD resolution videos, which disrupts synchronization between the two cameras.

## A.2 DATASET DETAILS AND STATISTICS

This section provides detailed information about the UDC-VIX dataset.

**Statistics.** From a pool of 647 videos, we have randomly selected 510 for training, 69 for validation, and 68 for the test set. The UDC-VIX dataset will be available in `PNG` format accompanied by a conversion script from `PNG` to `NPY`. We offer the dataset in `MP4` format for the review process to facilitate video quality assessment. Moreover, we have also annotated each video pair, providing a detailed overview of the total count and the distribution of different annotation labels. The video pairs are thoughtfully categorized into various settings, including the presence of flare and light sources, human presence and types of human motion, and indoor/outdoor.

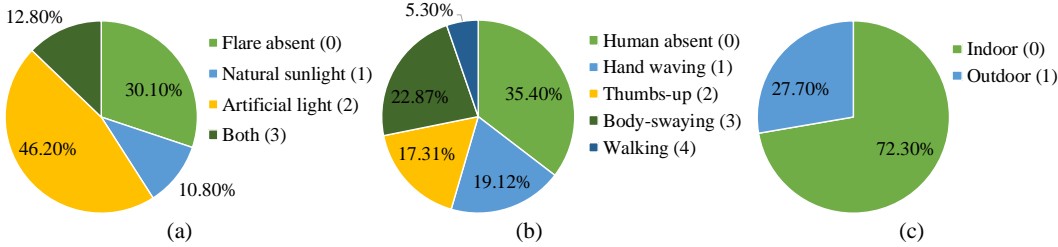

Figure A.2: The dataset distribution. The parenthesis beside a label is the encoding of the label. Note that a video pair can have multiple annotation labels. (a) The distribution of the lighting conditions. (b) The distribution of the human's presence and their actions. (c) The distribution of the shooting location.

**IRB approval.** We have obtained Institutional Review Board (IRB) approval for our UDC-VIX dataset, as our research involves human subjects. This rigorous process ensures the highest standards of research ethics. Using IRB-approved procedures, we enlisted 22 voluntary research participants. As shown in Table A.1, the IRB-approved participant information sheet provides comprehensive instructions verbally explained on the shoot day.

Similarly, it is essential to note that the users of the UDC-VIX dataset are engaged in research involving human subjects. Therefore, the users are required to secure IRB approval by the regulations

Table A.1: Prescribed instructions from IRB-approved participant information sheet. Out of thirty shots per person, videos displaying issues such as being out-of-focus are eliminated from the dataset.

**Q. What procedures will be followed if the participants take part in the study?**

**A.** If the participants agree to take part in, the following procedures will be conducted:
The participants will be photographed with 30 shots using the UDC and regular digital cameras according to the following motions:

- 5-second shots of body-swaying × 9 shots (6 indoors / 3 outdoors)
- 5-second shots of waving hands × 9 shots (6 indoors / 3 outdoors)
- 5-second shots of giving a thumbs-up × 9 shots (6 indoors / 3 outdoors)
- 5-second shots of walking indoors/outdoors × 3 shots

Since the UDC camera is located under the display and operates in low-light environments, it is necessary to shoot in various locations (indoors/outdoors) and conditions (bright/dark) to reflect the diverse quality degradation patterns of the UDC. Additionally, it is crucial to recognize individuals from various angles for tasks like face recognition, especially for personal authentication in the financial sector. Therefore, we must develop deep-learning models that restore the subject's appearance from different angles (e.g., front, left, and right), necessitating a dataset with shots from various angles. The recorded videos will be publicly released as a dataset for the UDC research.

**Q. How long will the study participation last?**

**A.** The study will take approximately 30 minutes. While the actual recording will take 2 minutes and 30 seconds (5 seconds × 30 shots), additional time will be needed for:

- The subject's shooting angle adjustments (5 minutes)
- Moving between locations (5 minutes)
- Checking alignment accuracy after moving (5 minutes)
- Making necessary adjustments (10 minutes)

**Q. Will compensation be provided for participating in this study?**

**A.** As a token of gratitude for participating in the study, the participants will receive a Starbucks gift card worth 50,000 Korean won. However, suppose the participants withdraw from participation before completing the 30 shots or request the disposal of the captured videos. In that case, we regret to inform the participants that compensation cannot be provided. Compensation will be provided to those who assist in fully completing the 30-shot video capture. Should the participants request the disposal of the videos after compensation has been provided, they will be required to return the compensation amount.

of their respective countries. When the users download the dataset, there will be instructions about the IRB approval, as shown in Figure A.3.

A.3   RIGOROUS MAINTENANCE PLAN

This section provides the UDC-VIX's easy accessibility and rigorous maintenance plan for long-term preservation.

**Easy accessibility.**   The UDC-VIX dataset will be publicly available at our research group's homepage (accessible in the camera-ready) as depicted in Figure A.3, improving accessibility.

Users can access the dataset by filling out a form on the research group's homepage. Upon submission, they will receive an email with the download link. Instructions for accessing the UDC-VIX dataset will also be provided on our project site, guiding users to the research group's homepage for

Figure A.3: Our Research Group's homepage section for the UDC-VIX dataset, which offers information and download access. It is temporarily inaccessible during the review period and will be available in the camera-ready version.

download. Distributing the dataset via the research group's homepage ensures long-term preservation. Handling contact and bug reports via email allows for continuous maintenance and updates.

**License.** The UDC-VIX dataset is licensed under the Creative Commons Attribution-NonCommercial-ShareAlike 4.0 International (CC BY-NC-SA 4.0). Under this license, the users of the UDC-VIX dataset can freely utilize, share, and modify this work by adequately attributing the original author, distributing any derived works under the same license, and utilizing it exclusively for non-commercial purposes. It is essential to mention that the UDC-VIX dataset is restricted to UDC research purposes only, as outlined in our IRB documentation. Detailed information about this license can be found in the official Creative Commons website.

# B  ANALYSIS DETAILS

This section describes the novelty of the UDC-VIX dataset in two ways. One is to detail the limitations of a synthetic dataset (e.g., VidUDC33K (Liu et al., 2024)). The other is to offer experimental results using six benchmark models such as DISCNet (Feng et al., 2021), UDC-UNet (Liu et al., 2023), FastDVDNet (Tassano et al., 2020), EDVR (Wang et al., 2019), ESTRNN (Zhong et al., 2020), and DDRNet (Liu et al., 2024). We also provide training details to ensure reproducibility.

## B.1  REASONS OF THE STRANGE SCENES IN VIDUDC33K

In Section 4 in the main body of the paper, we describe less meaningful and strange scenes in the VidUDC33K dataset. Two main strange phenomena exist in the VidUDC33K dataset. One is *the flare appearance in improbable situations* and *unintended white artifacts*. The other is *the darkened and nearly featureless degraded frames*.

**Improbable situations and unintended white artifacts.**    Liu et al. (2024) endeavor to synthesize flares through the convolution of the PSF with ground-truth images. However, the desired flares do not manifest as expected. Subsequently, they employ a scaling procedure to pixel values exceeding a certain threshold to amplify those values, which is followed by PSF convolution. This results in the flare appearance in *improbable situations* and *unintended white artifacts*. Flares in improbable scenarios are described in Figure 8 in the main body of the paper and Figure B.1. As for unintended

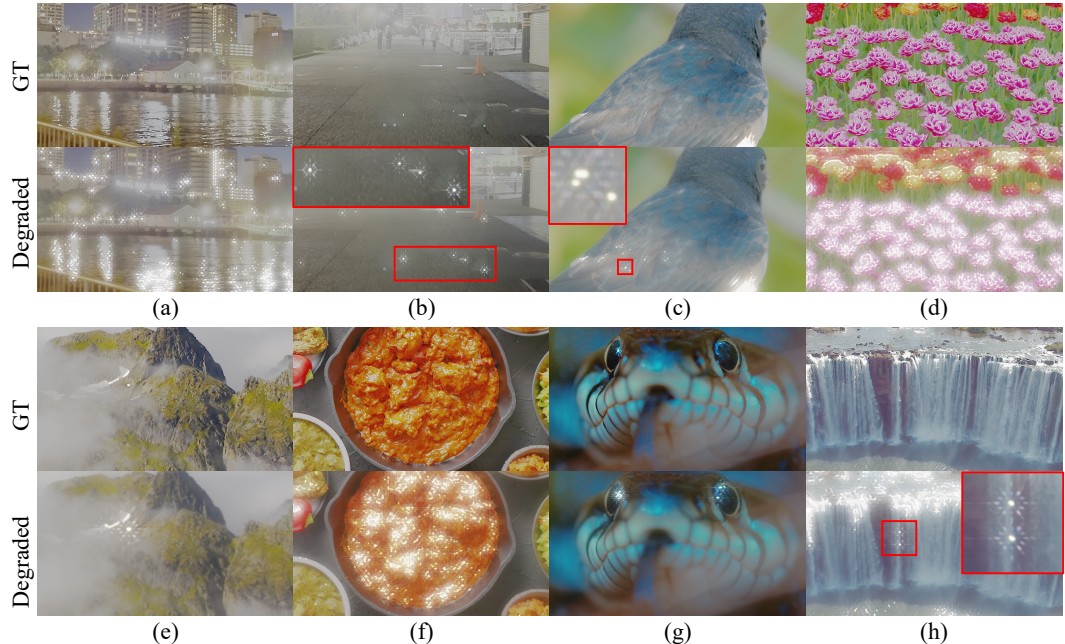

Figure B.1: The visual illustration that showcases improbable flares resulting from excessive scaling in the VidUDC33K dataset (Liu et al., 2024). (a) Flare in the river. (b) Flare from the dust on the camera lens. (c) Flare on the bird feathers. (d) Flare on the flower petals. (e) Flare on the mountain peaks. (f) Flare on the food. (g) Flare in the snake eyes. (h) Flare on the waterfalls.

white artifacts, pixel values exceeding a certain threshold are amplified, resulting in artifacts in regions close to white. Consequently, areas with clouds in the sky, waterfalls, and white walls become excessively white, losing their original color, as depicted in Figure B.2. Experiments are conducted without applying scaling to verify that the scaling is related to the flare generation. The results presented in the final row of Figure B.2 demonstrate that without scaling, flares do not manifest even in frames where they are expected to appear. Approximately 12% of the videos exhibit unintended white artifacts due to the scaling procedure, which is unsuitable for deep learning training.

**The darkened and nearly featureless frames.** Liu et al. (2024) strive to create temporally variant flares in continuous video sequences. They simulate the dynamic changes of the PSF during motion by computing the inter-frame homography matrix $H_{t-1 \to t}$, formulated as Equation 3, between consecutive frames.

$$
\begin{aligned}
k_t &= \mathcal{T}(k_{t-1}, H_{t-1 \to t}) \\
&= \left| \mathcal{F} \left( H_{t-1 \to t}^{-1} \left( \mathcal{F}^{-1} \left( \sqrt{k_{t-1}} \right) \right) \right) \right|^2, \\
H_{t-1 \to t} &= \mathcal{M}(I_{t-1}^{GT}, I_t^{GT}),
\end{aligned}
\tag{3}
$$

where $\mathcal{T}(\cdot)$ is the transformation function that utilizes $H_{t-1 \to t}^{-1}$ to perform a perspective warp on the PSF of the previous frame, $k_{t-1}$. $H_{t-1 \to t}^{-1}$ denotes the inverse matrix of $H_{t-1 \to t}$. $\mathcal{F}(\cdot)$ and $\mathcal{F}^{-1}(\cdot)$ represent the Fourier transform and its inverse, respectively. $\mathcal{M}(\cdot)$ is the matching component used to calculate the homography matrix between frames.

However, this process occasionally results in PSF values approaching zero, causing the degraded frames to appear entirely black. Specifically, this issue occurs in 4 out of 677 videos, as depicted in Figure B.3. The first frame does not undergo PSF transformation, while subsequent frames do. Therefore, as seen in Figure B.3(c), only the frames after the first one (e.g., the tenth frame) sometimes become black.

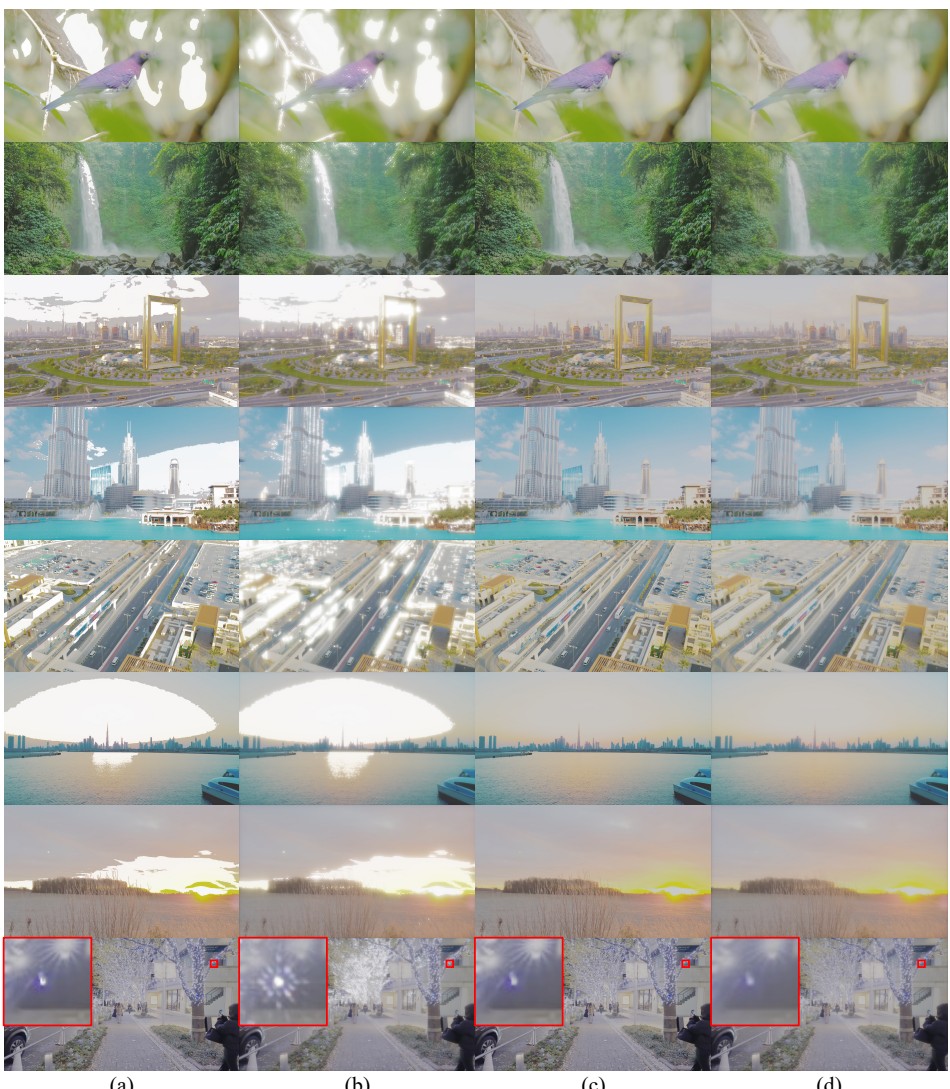

(a)  (b)  (c)  (d)

Figure B.2: The visual depiction that shows white artifacts resulting from excessive scaling in the VidUDC33K dataset (Liu et al., 2024). The frames *without* the scaling procedure do not exhibit these white artifacts, unlike the frames *with* the scaling procedure. Additionally, the flares in the frames with the scaling procedure are not visible in the frames without the scaling procedure. It appears that the authors use scaling to generate flares, inadvertently creating unrealistic white artifacts in the process. (a) The ground-truth frame *with* scaling procedure. (b) The degraded frame *with* scaling procedure. (c) The ground-truth frame *without* scaling procedure. (d) The degraded frame *without* scaling procedure.

## B.2 QUANTITATIVE RESULTS OF THE BENCHMARK MODELS

Among the categories illustrated in Figure A.2, the light conditions and shooting location are related to the restoration performance. In Table B.1, although the presence of humans seems to influence restoration performance, it is not directly correlated. To ensure the safety of participants, 86.4% of scenes, including humans, are captured indoors, which causes less severe degradation than outdoor natural flares. Given that a UDC-VIX video can have multiple annotations (e.g., an outdoor scene with flares caused by natural sunlight), the annotation type listed in a column in Table B.1 cannot be considered the only factor influencing UDC degradation. However, it is reasonable to recognize the annotation type as a significant factor affecting PSNR, SSIM, and LPIPS values.

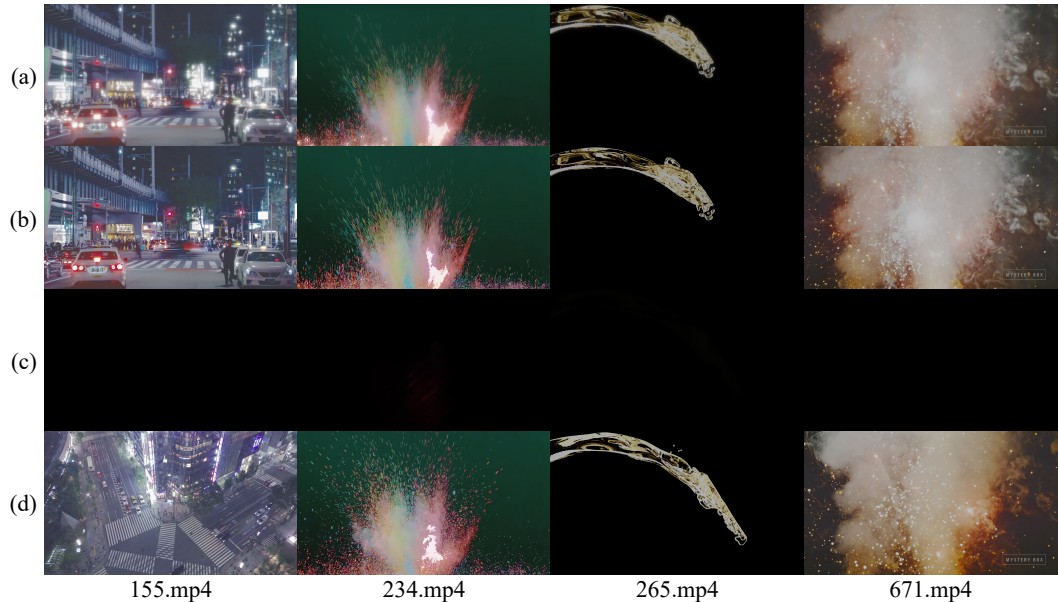

(a)

(b)

(c)

(d)

| 155.mp4 | 234.mp4 | 265.mp4 | 671.mp4 |

Figure B.3: The visual representation that demonstrates black frames resulting from incorrectly transformed PSFs in the VidUDC33K dataset Liu et al. (2024). (a) The first frame of the degraded video. (b) The first frame of the ground-truth video. (c) The tenth frame of the degraded video. (d) The tenth frame of the ground-truth video.

**Light sources.** As shown in Table B.1, all models encounter difficulties in restoring scenes with flare (Flare - Present - Average) compared to those without flare (Flare - Absent). Within flare-present scenes, the severity of degradation varies based on the light source (e.g., natural sunlight, artificial light, or both). Intense sunlight can oversaturate pixel values, obscuring objects around the flares (see Figure B.4(a) and (c)). Consequently, the benchmark models have more difficulty restoring videos with flares caused by natural sunlight than those caused by artificial light.

**Shooting location.** The benchmark models struggle more with restoring outdoor scenes than indoor scenes, as shown in Table B.1. Approximately 33.3% of outdoor and 15.4% of indoor scenes include flares caused by natural sunlight in the UDC-VIX dataset. Moreover, sunlight-induced flares occurring indoors are often less severe than those occurring outdoors. For example, outdoor scenes with natural sunlight flares show severe flare, as shown in Figure B.4(a), whereas indoor scenes with the same type of flares tend to be less severe, as depicted in Figure B.4(c) and (d). Notably, the flare in the upper right corner of Figure B.4(d) is mild, a result of sunlight scattered by a glass window rather than entering the camera directly. This understanding is crucial as it highlights the unique challenges of restoring outdoor scenes where direct sunlight is a significant factor. Consequently, all models face more significant difficulties restoring outdoor scenes than indoor scenes.

### B.3 QUALITATIVE RESULTS OF THE BENCHMARK MODELS

This section presents the visual results of the restored frames. The restoration outputs from benchmark models, which highlight various degradations that these models have yet to address, demonstrate the novelty of the UDC-VIX dataset and emphasize the importance of developing deep-learning models using real-world dataset.

**Light sources.** As illustrated in Figure B.4, flares can be categorized into glare, shimmer, and streak (Ahn et al., 2024; Dai et al., 2022). A glare is characterized by intense and robust light, resulting in circular patterns as artifacts. Shimmer entails rapid and nuanced light or color intensity variations across an image. A streak manifests as a lengthy, slender, and usually irregular line of light or color within an image.

Table B.1: Comparison of restoration performance. Each row's best and worst scores within each category are bold-faced and underlined, respectively.

| Model | Metric | Flare presence and light sources | | | | | Shooting location | | Human presence | | Average |
|---|---|---|---|---|---|---|---|---|---|---|---|
| | | Present | | | | Absent | Indoor | Outdoor | Present | Absent | |
| | | Natural sunlight | Artificial light | Both | Average | | | | | | |
| DISCNet (Feng et al., 2021) | PSNR ↑ | 21.93 | 24.15 | 22.37 | 23.55 | 26.53 | 25.43 | 21.92 | 26.94 | 20.18 | 24.53 |
| | SSIM ↑ | 0.7495 | 0.8451 | 0.8191 | 0.8255 | 0.8546 | 0.8573 | 0.7708 | 0.8795 | 0.7550 | 0.8351 |
| | LPIPS ↓ | 0.2925 | 0.2894 | 0.3250 | 0.2945 | 0.2206 | 0.2608 | 0.2973 | 0.2247 | 0.3521 | 0.2702 |
| UDC-UNet (Liu et al., 2023) | PSNR ↑ | 23.20 | 27.76 | 25.36 | 26.67 | 29.91 | 29.13 | 23.71 | 31.34 | 21.25 | 27.74 |
| | SSIM ↑ | 0.7962 | 0.8995 | 0.8857 | 0.8802 | 0.8954 | 0.9092 | 0.8158 | 0.9276 | 0.8088 | 0.8852 |
| | LPIPS ↓ | 0.2167 | 0.1814 | 0.2173 | 0.1920 | 0.1596 | 0.1679 | 0.2204 | 0.1398 | 0.2563 | 0.1814 |
| FastDVDNet (Tassano et al., 2020) | PSNR ↑ | 22.80 | 23.78 | 21.49 | 23.32 | 24.67 | 24.34 | 22.10 | 25.34 | 20.92 | 23.76 |
| | SSIM ↑ | 0.7696 | 0.8523 | 0.8245 | 0.8347 | 0.8474 | 0.8593 | 0.7798 | 0.8720 | 0.7792 | 0.8388 |
| | LPIPS ↓ | 0.2927 | 0.2772 | 0.3048 | 0.2834 | 0.2414 | 0.2568 | 0.3065 | 0.2364 | 0.3294 | 0.2696 |
| EDVR (Wang et al., 2019) | PSNR ↑ | 21.54 | 23.14 | 21.58 | 22.67 | 24.89 | 24.07 | 21.47 | 25.11 | 20.32 | 23.40 |
| | SSIM ↑ | 0.7515 | 0.8422 | 0.8132 | 0.8231 | 0.8380 | 0.8484 | 0.7690 | 0.8612 | 0.7682 | 0.8280 |
| | LPIPS ↓ | 0.2836 | 0.2843 | 0.3039 | 0.2867 | 0.2359 | 0.2605 | 0.2975 | 0.2390 | 0.3259 | 0.2700 |
| ESTRNN (Zhong et al., 2020) | PSNR ↑ | 22.99 | 25.54 | 24.08 | 24.92 | 25.70 | 26.07 | 22.60 | 26.99 | 21.91 | 25.18 |
| | SSIM ↑ | 0.7805 | 0.8818 | 0.8577 | 0.8615 | 0.8567 | 0.8847 | 0.7884 | 0.8938 | 0.7990 | 0.8599 |
| | LPIPS ↓ | 0.2670 | 0.2192 | 0.2640 | 0.2331 | 0.2087 | 0.2086 | 0.2725 | 0.1920 | 0.2845 | 0.2251 |
| DDRNet (Liu et al., 2024) | PSNR ↑ | 22.61 | 24.14 | 23.49 | 23.80 | 25.89 | 25.35 | 22.00 | 26.43 | 20.98 | 24.49 |
| | SSIM ↑ | 0.7799 | 0.8628 | 0.8455 | 0.8465 | 0.8524 | 0.8697 | 0.7870 | 0.8810 | 0.7898 | 0.8484 |
| | LPIPS ↓ | 0.2578 | 0.2267 | 0.2434 | 0.2341 | 0.2079 | 0.2079 | 0.2765 | 0.1936 | 0.2830 | 0.2255 |

As outlined in Section 4 in the main body of the paper, flares differ based on the light sources (i.e., *light source variant flare*). Additionally, even with the same light source, flares vary depending on the location (i.e., *spatially variant flare*). In Figure B.4(a) and (c), sunlight-induced flares are intense, causing all models to struggle to restore obscured objects. Conversely, artificial light in Figure B.4(b) and (c) is relatively easier to restore than sunlight-induced flares. However, benchmark models still face challenges restoring areas affected by shimmer and streak, resulting in speckled artifacts around the flare edges. The mild flare caused by natural light in Figure B.4(d) originates from sunlight scattered by a glass window, which all models restore well. Light sources like the one shown in Figure B.4(e), covered by a diffuser, produce less severe flares, leading to effective restoration by all models. As shown in Figure B.4(f), deep-learning models restore the glare and shimmer of fluorescent light, though the restoration of the blurred flare on the human face varies among models.

**Shooting location.** The visual restoration performance is sometimes influenced by the presence or absence of flares within the frame rather than solely by the shooting location. For instance, while Figure B.4(b) and (h) portray indoor scenes, models generally excel in restoring the flare-free frame in Figure B.4(h). Likewise, in outdoor scenes depicted in Figure B.4(a) and (i), models tend to achieve better restoration for the flare-free frame in Figure B.4(i). However, it is worth noting that some models may inaccurately render the sky with a reddish hue.

**Human.** The presence of humans alone does not pose a significant challenge to restoration. Instead, the restoration difficulty hinges on how UDC degradations, such as noise, blur, transmittance decrease, and flare, impact humans. For example, in Figure B.4(d) and (e), despite the presence of flares in the frames, they do not affect humans. However, in Figure B.4(f), the reflection of fluorescent light on the person's glasses poses challenges for restoring fine details around the eyes. In Figure B.4(g) and (h), human faces appear reddish in the input frames compared to the ground-truth frames due to UDC-induced diffraction occurring differently across RGB channels. Moreover, the restored facial colors vary among models. In applications like face recognition for smartphone unlocking, financial authentication, and video conferencing, it is crucial to consider these diverse UDC degradations for accurate human restoration since facial color is crucial in images or videos.

**Flicker.** The visual comparison in the paper can only show a single frame. Despite some successful restoration results of a frame in Figure B.4, multiple frames in the video often exhibit flickering across all models. This flickering may result from varying degradations between consecutive frames, such as transmittance decreases and flares. To see the flickering of the restored videos, please visit our project site.

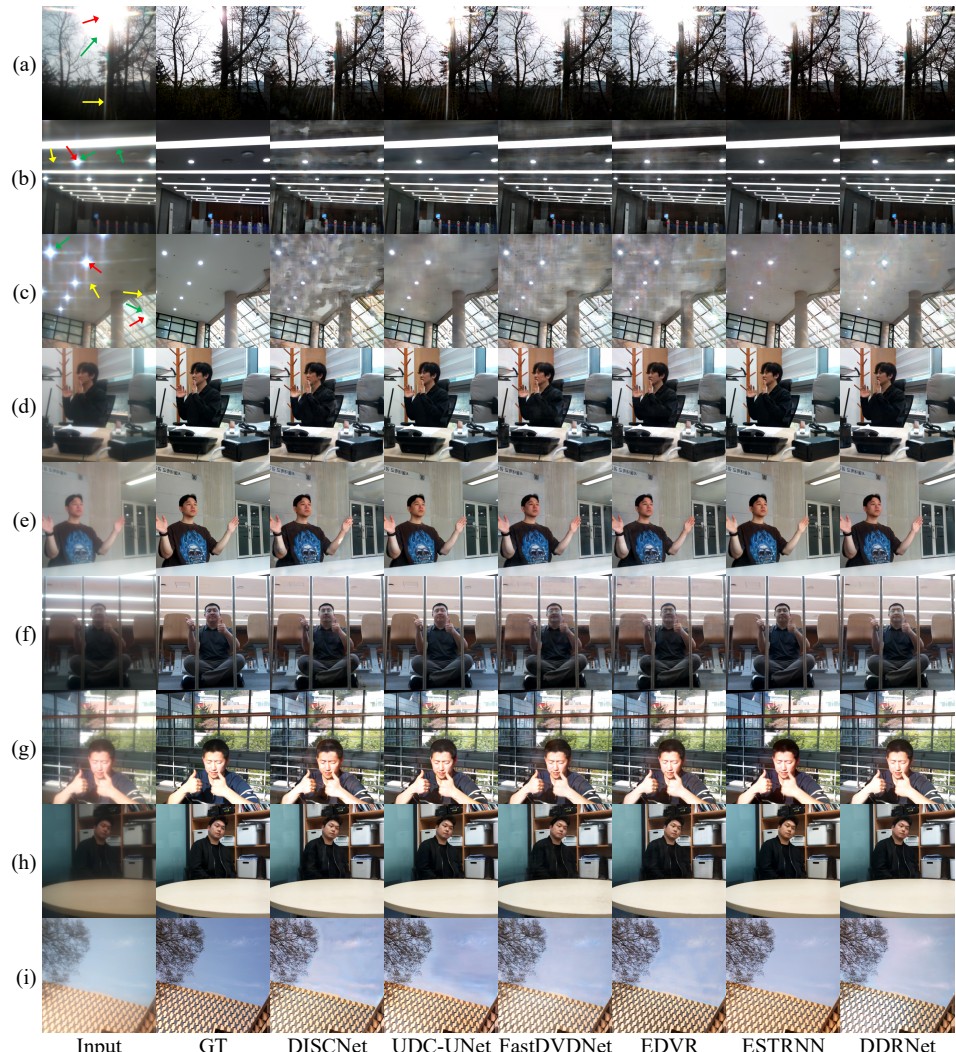

Figure B.4: The visual comparison of the restoration performance regarding different annotations. The red, green, and yellow arrows represent the flares' glare, shimmer, and streak, respectively. (a) Natural sunlight + Human absent + Outdoor. (b) Artificial light + Human absent + Indoor. (c) Both + Human absent + Indoor. (d) Natural sunlight + Hand waving + Indoor. (e) Artificial light + Hand waving + Indoor. (f) Artificial light + Thumbs-up + Indoor. (g) Natural sunlight + Thumbs-up + Indoor. (h) Flare absent + Body-swaying + Indoor. (i) Flare absent + Human absent + Outdoor.

## B.4 CROSS-DATASET VALIDATION

This section demonstrates the cross-dataset validation to tackle the unique dataset distribution and degradation patterns of UDC datasets as discussed in Section 6. For example, Samsung Galaxy Z-Fold 3 (Samsung Electronics Co., Ltd., 2021) (UDC-SIT (Ahn et al., 2024)) and Samsung Galaxy Z-Fold 5 (Samsung Electronics Co., Ltd., 2023) (UDC-VIX) share similar pixel designs, they still exhibit differences. Similarly, Samsung Galaxy Z-Fold 5 (UDC-VIX) and ZTE Axon 20 (ZTE Corporation, 2020) (VidUDC33K (Liu et al., 2024)) have vastly different pixel designs, as they come from different vendors.

Figure B.5(a) and (c) illustrate that the UDC-SIT and UDC-VIX datasets show similar degradation, such as blur, transmittance decrease, and flare shape. In contrast, Figure B.5(b) and (c) highlight the stark difference between the VidUDC33K and UDC-VIX datasets. This discrepancy arises from

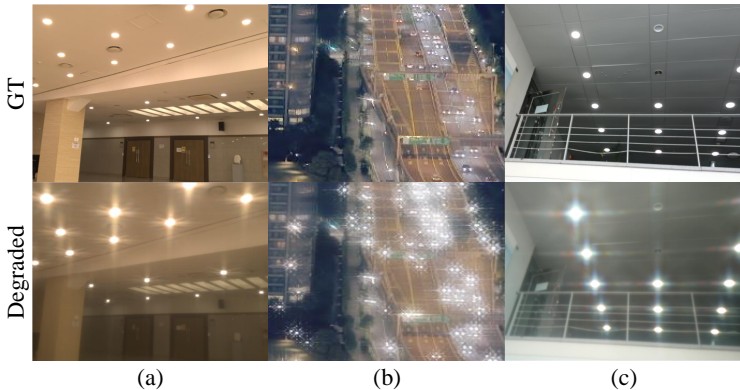

Figure B.5: Comparison of the UDC datasets, showing varied data distribution and degradation patterns. (a) UDC-SIT (Ahn et al., 2024). (b) VidUDC33K (Liu et al., 2024). (c) UDC-VIX. The first and second rows showcase GT and UDC-degraded, respectively.

Table B.2: The design of experiments demonstrating the effect of fine-tuning and the the use of a real-world dataset (e.g., UDC-VIX). The first and the second subscripts beside $\mathcal{M}$ indicate the training and fine-tuning datasets, respectively. For example, $\mathcal{M}_{s3}$ refers to the model trained on UDC-SIT without fine-tuning, while $\mathcal{M}_{s3s5}$ denotes the model trained on UDC-SIT and subsequently fine-tuned on UDC-VIX. Models without subscripts are trained and tested on the same dataset.

| Experiments | Model name | Training dataset | Fine-tuning dataset | Test dataset |
|---|---|---|---|---|
| | $\mathcal{M}_{s3}$ | UDC-SIT | - | UDC-VIX |
| Exp. 1 | $\mathcal{M}_{s3s5}$ | UDC-SIT | UDC-VIX | UDC-VIX |
| | $\mathcal{M}$ | UDC-VIX | - | UDC-VIX |
| | $\mathcal{M}_{s5}$ | UDC-VIX | - | VidUDC33K |
| Exp. 2 | $\mathcal{M}_{s5z20}$ | UDC-VIX | VidUDC33K | VidUDC33K |
| | $\mathcal{M}$ | VidUDC33K | - | VidUDC33K |
| | $\mathcal{M}_{z20}$ | VidUDC33K | - | UDC-VIX |
| Exp. 3 | $\mathcal{M}_{z20s5}$ | VidUDC33K | UDC-VIX | UDC-VIX |
| | $\mathcal{M}$ | UDC-VIX | - | UDC-VIX |

two factors: the variation in pixel design and the synthetic nature of the VidUDC33K dataset, which results in unrealistic degradation patterns.

Fine-tuning models to address variant dataset distributions or degradation patterns is crucial in practical applications. To evelute the effect of fine-tuning and validate the effectiveness of UDC-VIX, which reflects real-world degradation, we conduct three experiments (Exp. 1-3), as shown in Table B.2. The model names with or without subscripts specify the datasets used for training, fine-tuning, and testing. For example, $\mathcal{M}_{s3}$ refers to the model trained on UDC-SIT (**S**amsung Galaxy Z-Fold **3**) without fine-tuning, $\mathcal{M}_{s5z20}$ is trained on UDC-VIX (**S**amsung Galaxy Z-Fold **5**) and fine-tuned on VidUDC33K (**Z**TE Axon **20**), while $\mathcal{M}_{z20s5}$ is trained on VidUDC33K (**Z**TE Axon **20**) and fine-tuned on UDC-VIX (or **S**amsung Galaxy Z-Fold **5**). We use models $\mathcal{M}$ such as UDC-UNet (Liu et al., 2022b), DISCNet (Feng et al., 2021), and DDRNet (Liu et al., 2024) among six benchmark models in Table 3, given computational resource constraints. Fine-tuning is performed for 10% or 20% of the total iterations, with the learning rate set to 10% or 20% of the original value.

**Experiment 1: impact of fine tuning on UDC-VIX.** This experiment evaluates the impact of fine-tuning on UDC-VIX by comparing the performance of models trained on UDC-SIT when tested

Table B.3: **[Exp. 1]** Restoration performance of DISCNet (Feng et al., 2021) and UDC-UNet (Liu et al., 2022b) trained on UDC-SIT (Ahn et al., 2024), with and without additional fine-tuning on UDC-VIX. Models without subscripts refer to those trained and tested on UDC-VIX without fine-tuning, as detailed in Table 3. The number of iterations represents the percentage of fine-tuning iterations relative to the total iterations in the original configurations the authors provide.

| Model name | PSNR ↑ | SSIM ↑ | LPIPS ↓ | Training | Fine-tuning (# Iterations) | Test |
|---|---|---|---|---|---|---|
| DISCNet$_{s3}$ | 16.83 | 0.7107 | 0.3307 | UDC-SIT | - | UDC-VIX |
| DISCNet$_{s3s5}$ | 23.03 | 0.8231 | 0.2550 | UDC-SIT | UDC-VIX (10%) | UDC-VIX |
| DISCNet$_{s3s5}$ | 23.43 | 0.8280 | 0.2483 | UDC-SIT | UDC-VIX (20%) | UDC-VIX |
| DISCNet | 24.53 | 0.8351 | 0.2702 | UDC-VIX | - | UDC-VIX |
| UDC-UNet$_{s3}$ | 17.24 | 0.7228 | 0.3409 | UDC-SIT | - | UDC-VIX |
| UDC-UNet$_{s3s5}$ | 24.77 | 0.8656 | 0.2145 | UDC-SIT | UDC-VIX (10%) | UDC-VIX |
| UDC-UNet$_{s3s5}$ | 25.23 | 0.8703 | 0.2046 | UDC-SIT | UDC-VIX (20%) | UDC-VIX |
| UDC-UNet | 27.74 | 0.8852 | 0.1814 | UDC-VIX | - | UDC-VIX |

Table B.4: **[Exp. 2]** Restoration performance of DISCNet (Feng et al., 2021), UDC-UNet (Liu et al., 2022b), and DDRNet (Liu et al., 2024) trained on UDC-VIX, with and without additional fine-tuning on VidUDC33K (Liu et al., 2024). Models without subscripts refer to those trained directly on VidUDC33K, as shown in Table 3. The number of iterations represents the percentage of fine-tuning iterations relative to the total iterations in the original configurations the authors provide.

| Model name | PSNR ↑ | SSIM ↑ | LPIPS ↓ | Training | Fine-tuning (# Iterations) | Test |
|---|---|---|---|---|---|---|
| DISCNet$_{s5}$ | 18.73 | 0.7503 | 0.4159 | UDC-VIX | - | VidUDC33K |
| DISCNet$_{s5z20}$ | 28.89 | 0.9129 | 0.1727 | UDC-VIX | VidUDC33K (10%) | VidUDC33K |
| DISCNet | 28.89 | 0.8405 | 0.2432 | VidUDC33K | - | VidUDC33K |
| UDC-UNet$_{s5}$ | 19.84 | 0.7682 | 0.3737 | UDC-VIX | - | VidUDC33K |
| UDC-UNet$_{s5z20}$ | 29.57 | 0.9139 | 0.1506 | UDC-VIX | VidUDC33K (10%) | VidUDC33K |
| UDC-UNet | 28.37 | 0.8361 | 0.2561 | VidUDC33K | - | VidUDC33K |
| DDRNet$_{s5}$ | 20.10 | 0.8313 | 0.3446 | UDC-VIX | - | VidUDC33K |
| DDRNet$_{s5z20}$ | 29.12 | 0.8994 | 0.2180 | UDC-VIX | VidUDC33K (10%) | VidUDC33K |
| DDRNet | 31.91 | 0.9313 | 0.1306 | VidUDC33K | - | VidUDC33K |

on UDC-VIX, with and without fine-tuning on UDC-VIX. For $\mathcal{M}_{s3}$ and $\mathcal{M}_{s3s5}$, we use UDC-UNet and DISCNet, two restoration models specifically designed for UDC still image, since UDC-SIT is the still image dataset. As presented in Table B.3, DISCNet$_{s3}$ and UDC-UNet$_{s3}$ trained exclusively on UDC-SIT struggle to generalize to UDC-VIX. In contrast, DISCNet$_{s3s5}$ and UDC-UNet$_{s3s5}$, which incorporate fine-tuning with UDC-VIX, demonstrate superior restoration performance for UDC-VIX degradations. Notably, increasing the number of fine-tuning iterations further enhances the performance.

These findings lead to the following conclusions: while Samsung Galaxy Z-Fold 3 (UDC-SIT) and Samsung Galaxy Z-Fold 5 (UDC-VIX) share similar pixel designs due to their origin from the same vendor, their differences are substantial enough to require fine-tuning. With adequate adaptation, however, these models effectively leverage degradations from other UDC devices, underscoring the potential for cross-device generalization with fine-tuning.

**Experiment 2: impact of fine tuning on VidUDC33K.** This experiment aims to assess the impact of fine-tuning on VidUDC33K. It compares the performance of models trained on UDC-VIX when tested on VidUDC33K, both with and without fine-tuning on VidUDC33K. For the models $\mathcal{M}_{s5}$ and $\mathcal{M}_{s5z20}$, we use UDC-UNet, DISCNet, and DDRNet, which are explicitly designed to address UDC degradations. These models, selected from the six benchmark models in Table 3, demonstrate the effectiveness of fine-tuning across datasets, even when the source and target de-

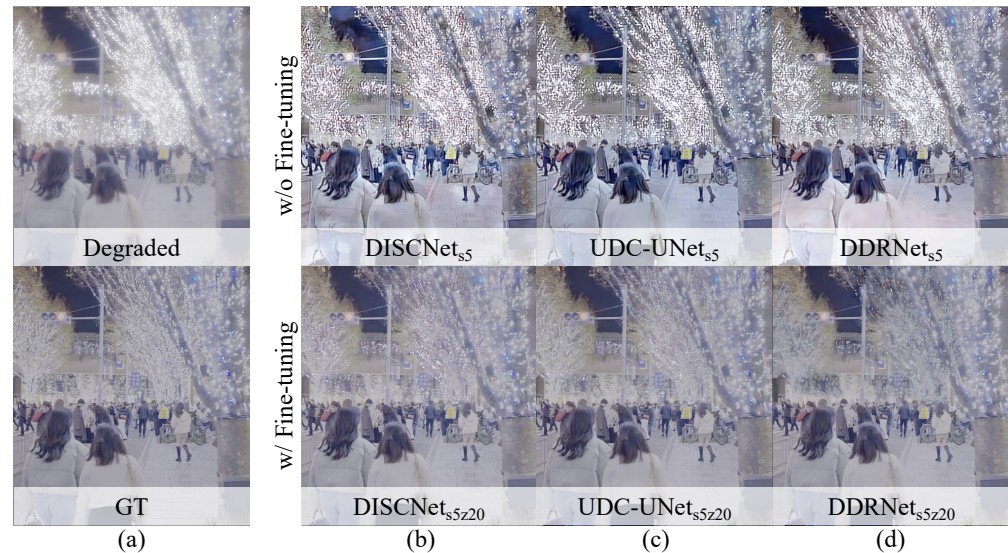

Figure B.6: **[Exp. 2]** Comparison of restoration performance across different models on the VidUDC33K dataset (Liu et al., 2024). (a) UDC-degraded (at first row) and GT (at second row) frames of VidUDC33K dataset. Test frames by the models such as (b) DISCNet, (c) UDC-UNet, and (d) DDRNet without (first row) and with (second row) fine-tuning. The models in the first row are pre-trained on UDC-VIX without fine-tuning on VidUDC33K. The models in the second row are pre-trained on UDC-VIX and fine-tuned on VidUDC33K, showing improved restoration performance.

vices differ, such as the Samsung Galaxy Z-Fold 5 and ZTE Axon 20. Notably, fine-tuning improves generalization and allows the models to perform well on different devices, as shown in Table B.4.

Interestingly, despite the differences in device architecture, the fine-tuned models DISCNet$_{s5z20}$ and UDC-UNet$_{s5z20}$ outperform DISCNet and UDC-UNet, solely trained by VidUDC33K, as shown in Table B.4. This performance boost can be attributed to the fact that UDC-VIX exhibits more realistic and severe degradation patterns such as noise, blur, transmittance decrease, and variant flares compared to the synthetic VidUDC33K dataset, as discussed in Section 4. Consequently, models pre-trained on UDC-VIX show improved performance with fine-tuning on VidUDC33K when tested on VidUDC33K, highlighting the benefits of using a real-world dataset.

Figure B.6 illustrates these findings. Models trained on UDC-VIX without fine-tuning (e.g., DISCNet$_{s5}$, UDC-UNet$_{s5}$, and DDRNet$_{s5}$) are able to restore blur but fail to address flare artifacts, as described in Figure B.6(c), (e), and (g). Interestingly, they show better restoration of transmittance decrease compared to VidUDC33K's ground truth, likely due to the brighter tone in UDC-VIX's ground truth compared to VidUDC33K's. On the other hand, models fine-tuned on VidUDC33K effectively restore the complex degradation patterns specific to VidUDC33K, underscoring the importance of pre-trained on real-world datasets like UDC-VIX, as shown in Figure B.6(d), (f), and (h).

**Experiment 3: comparison of UDC-VIX and VidUDC33K.** This experiment evaluates the effect of fine-tuning on UDC-VIX by comparing the performance of models trained on VidUDC33K when tested on UDC-VIX, with and without fine-tuning on UDC-VIX. For the models $\mathcal{M}_{z20}$ and $\mathcal{M}_{z20s5}$, we use DDRNet, which is the only publicly available pre-trained model trained on VidUDC33K. As shown in Table B.5, DDRNet$_{z20}$, when not fine-tuned on UDC-VIX, fails to effectively handle the complex, severe, and real-world degradations present in UDC-VIX. In contrast,

Table B.5: **[Exp. 3]** Restoration performance of DDRNet (Liu et al., 2024) trained by VidUDC33K (Liu et al., 2024), with and without additional fine-tuning on UDC-VIX. Models without subscripts refer to those trained directly on UDC-VIX, as shown in Table 3. The number of iterations represents the percentage of fine-tuning iterations relative to the total iterations in the original configurations the authors provide.

| Model name | PSNR ↑ | SSIM ↑ | LPIPS ↓ | Training | Fine-tuning (# Iterations) | Test |
|---|---|---|---|---|---|---|
| DDRNet$_{z20}$ | 11.34 | 0.5369 | 0.5584 | VidUDC33K | - | UDC-VIX |
| DDRNet$_{z20s5}$ | 21.79 | 0.8250 | 0.2560 | VidUDC33K | UDC-VIX (10%) | UDC-VIX |
| DDRNet | 24.49 | 0.8484 | 0.2255 | UDC-VIX | - | UDC-VIX |

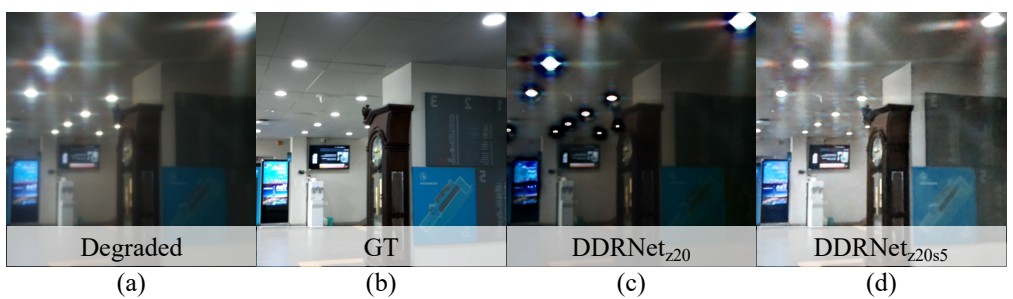

|  |  |  |  |
|---|---|---|---|
| Degraded | GT | DDRNet$_{z20}$ | DDRNet$_{z20s5}$ |
| (a) | (b) | (c) | (d) |

Figure B.7: **[Exp. 3]** Comparison of restoration performance across different models on the UDC-SIX dataset. (a) UDC-degraded and (b) GT images from the UDC-VIX dataset. Restored images by (c) DDRNet$_{z20}$, (d) DDRNet$_{z20s5}$. The model DDRNet$_{z20}$ is pre-trained on VidUDC33K without fine-tuning on UDC-VIX, while DDRNet$_{z20s5}$ is pre-trained on VidUDC33K and fine-tuned on UDC-VIX, showing improved restoration performance. However, compared to the results in Figure B.6, the fine-tuned model still struggles to handle the real-world degradations present in the UDC-VIX dataset, as it is originally trained on the synthetic VidUDC33K dataset.

DDRNet$_{z20s5}$, fine-tuned on UDC-VIX, demonstrates significant performance improvements over DDRNet$_{z20}$.

However, as illustrated in Figure B.7, even with fine-tuning, DDRNet$_{z20s5}$ still shows limitations in handling specific real-world degradations, such as severe flares. Unlike Experiment 2, where models are pre-trained on the real-world UDC-VIX dataset, Experiment 3, which involves pre-training on the synthetic VidUDC33K, highlights the challenges of leveraging realistic degradation patterns. These results emphasize pre-training models on real-world datasets like UDC-VIX to fully capture complex degradations that synthetic datasets cannot adequately represent.

## B.5 REPRODUCIBILITY

This section provides detailed information on the deep-learning models used to compare the UDC-VIX dataset in the paper for reproducibility. The code can be found and downloaded at our project site.

The learnable restoration models used for evaluating the UDC-VIX dataset include DISCNet (Feng et al., 2021), UDC-UNet (Liu et al., 2023), FastDVDNet (Tassano et al., 2020), EDVR (Wang et al., 2019), ESTRNN (Zhong et al., 2020), and DDRNet (Liu et al., 2024). We use a single-node GPU cluster to train each benchmark model. Each node has eight AMD Instinct MI100 GPUs. While we mainly stick to the original authors' code and training settings for the models, we introduce some modifications except ESTRNN.

- **DISCNet.** DISCNet is designed to restore UDC still images in high dynamic range (HDR) (e.g., SYNTH (Feng et al., 2021)). Accordingly, we modify the PyTorch DataLoader to use

normalization instead of Reinhard tone mapping (Reinhard et al., 2002). The DataLoader randomly selects one frame per video from the UDC-VIX dataset for each iteration during the training and validation phases.

- **UDC-UNet.** UDC-UNet is also designed to restore UDC still images in HDR. The original authors do not conduct normalization or tone mapping in the DataLoader and employ a tone mapping L1 loss function. However, since the UDC-VIX dataset has a low dynamic range (LDR), we modify the PyTorch DataLoader to use normalization. We clamp the model output between 0 and 1 and then calculate the L1 loss. The DataLoader randomly selects one frame per video from the UDC-VIX dataset for each iteration.

- **FastDVDNet.** FastDVDNet is a video denoising model that utilizes NVIDIA's Data Loading Library (DALI) (Nvidia, 2018), processing a noise map and multiple frames as inputs. Instead of DALI, we employ the PyTorch DataLoader tailored to the UDC-VIX dataset in `npy` format. We set the noise level to zero. To accommodate FHD resolution and multiple degradations in the UDC-VIX dataset, we increase the patch size from 64 to 256. Furthermore, we extend the training duration of FastDVDNet to 400 epochs, compared to the original 95, to ensure the model reaches full saturation.

- **EDVR.** To address out-of-memory issues with EDVR, which boasts 23.6 M parameters, we reduce the patch size from 256 to 192. Additionally, during inference on the test set, we divide it into two patches of size $3 \times 1,060 \times 1,060$ each and merge them afterward.

- **DDRNet.** During the inference process, the authors of DDRNet partition each frame into patches of size $3 \times 256 \times 256$ and input 50 frames simultaneously. However, patch-wise inference introduces the borderline between patches. To address this, we conduct inference at full resolution ($3 \times 1,060 \times 1,900$) with ten frames at a time.

## C    DISCUSSION ON THE RESPONSIBLE USE OF THE DATASET

This section discusses the potential negative societal impacts, the corresponding user guidelines, and our responsibility.

### C.1    POTENTIAL NEGATIVE SOCIETAL IMPACTS

The UDC-VIX dataset includes the faces and motions of 22 research participants, raising concerns about its potential for misuse, such as in deep fake applications. This technology can generate convincingly altered videos, threatening individual privacy and societal trust. Deep fakes can infringe upon personal integrity and privacy, leading to social unrest and confusion. Given these potential negative societal impacts, careful consideration is needed when using the dataset.

### C.2    USER GUIDELINES

The users of the UDC-VIX dataset are expected to adhere to the following guidelines:

- **Responsible use.** Users must ethically and responsibly utilize the dataset, ensuring it does not infringe on individual privacy or contribute to societal harm.

- **Compliance with legal and ethical standards.** Users must comply with all relevant legal and ethical standards, including obtaining Institutional Review Board (IRB) approvals by the regulations of their respective countries, and respect any restrictions or conditions imposed by the IRB or other regulatory bodies. Any violations of the laws of the Republic of Korea or the user's respective country will be the user's sole responsibility.

- **Restricted Usage.** Users must avoid using the UDC-VIX dataset for harmful applications, such as deep fake technologies or other misinformation or manipulation. Moreover, the 22 participants' agreed-upon research scope during our IRB review centers on acquiring UDC video datasets and developing restoration models. Therefore, this dataset must exclusively serve UDC research purposes.

## C.3 OUR RESPONSIBILITY

As custodians of the UDC-VIX dataset, we acknowledge our responsibility to:

- **Protect participant privacy.** Our foremost concern is preserving the privacy and confidentiality of research participants. While participants consented to the public use of their faces and motions within the dataset, we are dedicated to providing user guidance for appropriate research utilization and exerting efforts to safeguard other personal information.

- **Facilitate ethical use.** We provide comprehensive guidelines and documentation on *datasheets for datasets*, our project site, and our research group's homepage. The email automatically sends the download link when users complete the application form on our research group's homepage, which will also inform users about the dataset's potential risks and ethical considerations.

- **Respond to concerns.** Our commitment to the responsible management of the UDC-VIX dataset extends to promptly addressing any concerns or complaints raised. We value users' feedback and are ready to take appropriate actions, such as data corrections and updates, to mitigate potential harm or misuse if any misuse of the dataset is reported through our research group's homepage, as shown in Figure A.3.

