# OpenReview forum: "UDC-VIT: A Real-World Video Dataset for Under-Display Cameras"
_ICLR.cc/2025/Conference — Submitted to ICLR 2025_

### Official Review · Reviewer_NbRS · 2024-10-29

**Soundness:** 3
**Presentation:** 3
**Contribution:** 3
**Rating:** 8
**Confidence:** 4

**Summary:**

The authors provide a real-world UDC video dataset called UDC-VIX, which accurately reflects actual UDC video degradations. Besides, They compare UDC-VIX with an existing synthetic UDC video dataset using six deeplearning restoration models to demonstrate its effectiveness.

**Strengths:**

1. The authors provide a real-world UDC video dataset called UDC-VIX, which accurately reflects actual UDC video degradations.
2. The authors compare UDC-VIX with an existing synthetic UDC video dataset using six deeplearning restoration models to demonstrate its effectiveness.

**Weaknesses:**

1. In section 5.2 FACE RECOGNITION, the description of the experimental setup for face recognition is difficult to understand, but in theory, it should be well understood, so the author needs to sort out this part.
2. Even if it is not UDC, it will produce exaggerated flares under strong light. Another question is, does the rear camera of the iPhone belong to UDC?
3. Regarding the data distribution problem generated by specific devices mentioned in LIMITATIONS, since this problem exists, did the author solve it to some extent when proposing the data set, such as collecting data based on mobile phone brands with larger market share?
4. Does face recognition under UDC have practical application significance? Does UDC currently limit the performance of the face recognition algorithm? Are there any other problems?

**Questions:**

See Weaknesses for details.

---

### Official Review · Reviewer_ZDh7 · 2024-11-01

**Soundness:** 2
**Presentation:** 3
**Contribution:** 1
**Rating:** 5
**Confidence:** 4

**Summary:**

This work proposes a UDC video dataset UDC-VIX that contains realistic UDC degradations (e.g., low transmittance, blur, noise, and glare). Specifically, this work proposes an efficient video capture system to acquire a pair of matched UDC-degraded videos and ground truth videos through precise synchronization of two cameras. In addition, this work uses DFT to align UDC-VIX frame by frame, showing the highest alignment accuracy, which is sufficient for training deep learning models. Through comparative experiments, this work demonstrates the effectiveness of UDC-VIX.

**Strengths:**

1、This paper introduces the data set collection and processing process in detail, and the content is clear and concise.
2、This paper analyzes and compares the differences between existing data sets and the collected data sets, highlighting the necessity of creating new data sets.
3、This paper shows the results of the collected data sets in video reconstruction and face recognition, reflecting the effectiveness of this paper's data set.

**Weaknesses:**

1、There are some unclear introductions in this paper, such as the corresponding English abbreviations in Figure 1 are not introduced.
2、This paper focuses on describing the implementation details and does not reflect the innovation.
3、The experimental data in this paper is not sufficient to fully demonstrate the advantages of the dataset.

**Questions:**

1、This paper mentions that capturing paired videos through a beam splitter is an innovation of this work, but the method of capturing paired videos based on a beam splitter is not new, and there have been some works in other fields. In addition, capturing GT videos through a beam splitter will degrade because the amount of light is halved. Did the author consider this problem?

2、Fast-moving objects are excluded from the dataset, which has an impact on handling such situations. Compared with other datasets, does this dataset have a disadvantage in handling such videos, and how big is the disadvantage?

3、When visually comparing different datasets, different video frames are used for comparison, which is not convincing. It is recommended to use the same video frames for comparison.

4、This article emphasizes the superiority of the dataset, but lacks specific experimental results. For example, will the results of training on a new dataset be better when tested on other datasets?

5、This paper introduces the methods used in the data collection and processing process, but these methods are existing technologies and do not reflect the innovation of this paper.

6、This paper is more like an engineering implementation, and the innovation is not sufficient. Please rethink the innovation of this paper.

---

### Official Review · Reviewer_sGyh · 2024-11-03

**Soundness:** 3
**Presentation:** 3
**Contribution:** 2
**Rating:** 6
**Confidence:** 3

**Summary:**

Differing from unrealistic or synthetic UDC degradation in previous works, in this paper, the authors construct a video-capturing system to simultaneously acquire non-degraded and UDC-degraded videos of the same scene. The real-world UDC video dataset is collected and then aligned using DFT. Six representative methods are trained and tested on the dataset. Based on the UDC-VIX, the authors evaluate the image quality and face recognition accuracy of UDC restoration.

**Strengths:**

The objective of the paper is clear, the overall narrative is fairly complete, and the amount of work is substantial. According to the description in the paper, the collected dataset is more diverse in scenarios compared to previous datasets in the UDC field, and the types of degradation are also closer to real-world conditions.

**Weaknesses:**

1. The comparison of Table 3 shows the results of different methods trained and tested on two separate datasets and no cross-testing was conducted. This makes it difficult to assess the impact of different datasets on restoration methods.
2. As mentioned in LIMITATION, UDC degradations vary with the display pixel design, so which types of degradation will be affected? This requires more detailed explanation and analysis.

**Questions:**

Refer to Weaknesses.

---

### Official Review · Reviewer_NXxb · 2024-11-04

**Soundness:** 3
**Presentation:** 3
**Contribution:** 2
**Rating:** 5
**Confidence:** 4

**Summary:**

The paper proposes a video-capturing system that can simultaneously record UDC-degraded and ground-truth videos for the same scene. Applying this system and discrete Fourier transform, the authors collect an aligned real-world UDC video dataset called UDC-VIX that accurately represents real-world UDC video degradations. The paper demonstrates the effectiveness of UDC-VIX by comparing it with an existing synthetic UDC video dataset using six deep-learning restoration models.

**Strengths:**

This is the first real-world UDC video dataset that accurately represents real-world UDC video degradations.
The paper proposes a video-capturing system using a beam splitter to minimize discrepancies between paired frames, which is novel in the UDC field.
The paper provides cross-dataset validation experiments and the analysis of limitations of existing datasets such as unrealistic flare occurrences and white artifacts in the supplement.
The presentation of the paper is clear and organized.

**Weaknesses:**

The theoretical analysis of the limitations of existing datasets and the strength of the proposed new dataset would be better in the main paper instead of the appendix.

**Questions:**

1. This paper is more like a combination of existing techniques, the methods applied in data collection, alignment, and dataset evaluation are all not novel.
2. In Table 3, UDC-UNet achieves better SSIM and LPIPS scores on the UDC-VIX dataset while all other models exhibit decreased performance on the UDC-VIX dataset. Can you explain more about this phenomenon?
3. Due to the insufficient performance of existing models on your dataset, can you propose a better resolution to handle the real-world degradation in UDC videoes?

---

### Meta-Review · Area_Chair_GenH · 2024-12-20

**Metareview:**

Metareview: The paper presents a novel dataset, UDC-VIX, which is the first real-world video dataset associated with under-display cameras (UDC). The authors propose a video-capturing system that simultaneously acquires non-degraded and UDC-degraded videos of the same scene, aligning them frame by frame using discrete Fourier transform (DFT). The dataset is compared with existing synthetic UDC video datasets using six deep learning-based models, demonstrating the ineffectiveness of models trained on synthetic data in handling real-world UDC degradations.
This paper receives dispersed scores among the reviews. Reviewer NbRS and sGyh praised the dataset's effectiveness but suggested clarifying the experimental setup, addressing the data distribution problem, and adding cross-datasets validation. Reviewer NXxb appreciated the novelty of the dataset but noted the lack of theoretical contributions and suggested including more theoretical analysis in the main paper. Reviewer ZDh7 raised concerns about the innovation of the data collection methods and suggested further experiments to demonstrate the dataset's advantages.

**Additional Comments On Reviewer Discussion:**

The authors provided detailed responses to the reviewers' comments, including revisions to the paper to address the concerns. They added cross-dataset validation experiments, clarified the experimental setup for face recognition, and provided more detailed explanations of the dataset's limitations and strengths. Overall, the paper makes a valuable contribution to the field but the theoretical contributions and the novelty concerns are not addressed. Therefore, I recommend reject the paper.

---

### Decision · Program_Chairs · 2025-01-22

Reject